# Hamiltonian latent operators for content and motion disentanglement in image sequences

**Asif Khan**
School of Informatics
University of Edinburgh
asif.khan@ed.ac.uk

**Amos Storkey**
School of Informatics
University of Edinburgh
a.storkey@ed.ac.uk

## Abstract

We introduce *HALO* – a deep generative model utilising HAmiltonian Latent Operators to reliably disentangle content and motion information in image sequences. The *content* represents summary statistics of a sequence, and *motion* is a dynamic process that determines how information is expressed in any part of the sequence. By modelling the dynamics as a Hamiltonian motion, important desiderata are ensured: (1) the motion is reversible, (2) the symplectic, volume-preserving structure in phase space means paths are continuous and are not divergent in the latent space. Consequently, the nearness of sequence frames is realised by the nearness of their coordinates in the phase space, which proves valuable for disentanglement and long-term sequence generation. The sequence space is generally comprised of different types of dynamical motions. To ensure long-term separability and allow controlled generation, we associate every motion with a unique Hamiltonian that acts in its respective subspace. We demonstrate the utility of *HALO* by swapping the motion of a pair of sequences, controlled generation, and image rotations.

## 1   Introduction

The ability to learn to generate artificial image sequences has diverse uses, from animation, keyframe generation, and summarisation to restoration that has been explored in previous work over many decades (Hogg, 1983; Hurri and Hyvärinen, 2003; Cremers and Yuille, 2003; Storkey and Williams, 2003; Kannan et al., 2005). However, learning to generate arbitrary sequences is not enough; to hold a practical application, the user must be able to control aspects of the sequence generation, such as the motion being enacted or the characteristics of the agent doing an action. To enable this, we must learn to decompose image sequences into *content* and *motion* characteristics so that we can apply learnt motions to new objects or vary the motions being applied.

Deep generative models (DGMs) such as variational autoencoders (VAEs) (Kingma and Welling, 2013) and Generative Adversarial Networks (GANs) (Goodfellow et al., 2014) use neural networks (NNs) to transform the samples from a prior distribution over lower-dimensional latent factors to samples from the data distribution itself. Recent developments  (Chung et al., 2015; Srivastava et al., 2015; Hsu et al., 2017; Yingzhen and Mandt, 2018) extend VAEs to sequences using Recurrent Neural Networks (RNNs) on the representation of temporal frames. Similar approaches have been taken for GAN models (Tulyakov et al., 2018; Yoon et al., 2019; Dandi et al., 2020).

The dynamical processes creating the evolution of image sequences are highly constrained. Consider the simplistic case of a person walking in a scene with a camera moving around that individual. The walking pose will return to similar positions periodically, and likewise, the revolving camera will revisit previous positions. Even without strict periodicity, many dynamical processes are reversible. Any time that a dynamic could conceivably return to an earlier state suggests an implicit conservation law—the conservation of information in the underlying scene generator—as it must be capable of

36th Conference on Neural Information Processing Systems (NeurIPS 2022).

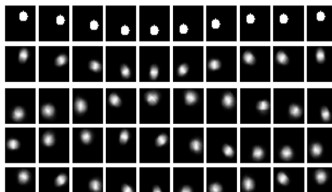 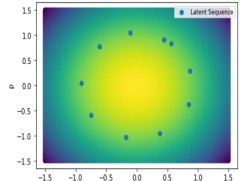 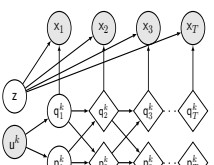 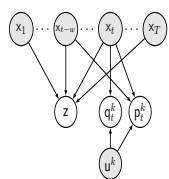

Figure 1: Left: we empirically demonstrate the benefits of learning Hamiltonian operator in the latent space. Consider $(32, 32)$ image sequence of a ball rotating in a fixed orbit; the $(i, j)$ centre of a ball moves under constraint $i^2 + j^2 = c$, where $c$ is a constant. We generated a data set of sequences with different initial condition and same number of time steps. Next, we trained a VAE with Hamiltonian operator in the latent space. We use an encoder to transform sequences to a 2-dimensional phase space, next we unroll a trajectory using a learnable Hamiltonian operator and finally use decoder to obtain the sequence. Top row is the original sequence, second row is the reconstructed sequence and below are three sequences generated from random initial states. Middle: the coordinates in the phase space are coloured by the energy value, alongwith a representation of a sequence generated from a random initial coordinate. Right: is the probabilistic graph of our generative and inference model.

returning to and regenerating the same scene with non-negligible probability. The critical observation we wish to capture in this paper is that understanding the conservation occurring in the context of a set of sequences is a vital ingredient to decomposing *content* from *motion*.

Given any conserved quantity, any motion must be modelled to maintain the conserved quantity. In physics, such a motion is called a *Hamiltonian* motion; it keeps the corresponding Hamiltonian function constant. Hence we argue that a flexible latent Hamiltonian model provides a good inductive bias to learn a representation space that enables conservation of the right quantities (which are themselves learnt) and models the dynamic evolution. This is mathematically equivalent to saying that we can learn to represent the underlying motions as combinations of differentiable symmetry groups; all differentiable symmetry transformations follow a conservation law (Noether, 1918).

We next illustrate the benefit of using Hamiltonian dynamics in the latent space of DGM in Figure 1.[1]. Here, we observe using the Hamiltonian dynamics model can discover constant energy in latent space from a set of image sequences proving critical for generating novel energy preserving sequences. This example demonstrates identifying symmetries is a suitable inductive bias for developing expressive DGMs that understand the motion constraints and generalise beyond the training data. Higgins et al. (2018); Toth et al. (2019); Botev et al. (2021) have discussed the benefits of such inductive biases for learning disentangled representation.

The existing sequential DGMs do not impose any structural prior for constraining the dynamics in motion space and, therefore, accumulate errors as the sequence length grows, quickly deviating from the relevant path (Karl et al., 2016; Fraccaro et al., 2017; Yildiz et al., 2019; Bird and Williams, 2019). The attractive property of Hamiltonian dynamics is that they are *symplectic* that is, the divergence of a vector field is zero, and the evolving dynamics preserve the infinitesimal volume element. Consequently, the motion paths are restricted to a low-dimensional manifold in the latent space, and we can predict the dynamics forward and backwards in time.

In this paper, we intimate the more general applicability of latent Hamiltonian models; previous applications have been limited to somewhat constrained physical systems. We propose *HALO* – a VAE framework to model the dynamics of image sequences using a collection of learnable linear Hamiltonian operators in the latent space. Specifically, for any motion sequence, we model the transition from a time step $t$ to a step $t + 1$ using a group action of a Hamiltonian operator. The evolution of the dynamics of a sequence leaves certain information unchanged, identified as *content*, and specific properties that evolve in conjunction (i.e. *motion*). Since the space of image sequences can comprise various types of dynamical actions, we split the *motion space* into subspaces where each subspace models a unique action and is unaffected by other actions. This formulation explicitly ensures the separability of dynamics. It further reduces the computational cost since the Hamiltonian of the space is now in a block diagonal form where each block is a Hamiltonian of a symmetry subgroup. Here, we focus on a discrete, identified set of actions that we can then compose at generation time. We want to remark our method can also work without action labels,

---

[1] We discuss the specifics of dynamical operators in Section 3

as empirically demonstrated in the results. The benefit of identifying actions apriori is that we can use it for a controlled generation. We empirically demonstrate the advantages of our approach through i) generation of diverse dynamics from a starting frame and ii) demonstrating successful disentanglement of the content and motion representation.

## 2   Related Work

**Hamiltonian Neural Networks** Several deep learning (DL) methods have recently been proposed to learn the dynamics of physical systems using Hamiltonian mechanics. Greydanus et al. (2019) use NNs to predict Hamiltonian from phase-space coordinates $\mathbf{s} = (\mathbf{p}, \mathbf{q})$ and their derivatives. In similar work (Bondesan and Lamacraft, 2019) used NNs to discover symmetries of Hamiltonian mechanical systems. More recently, Hamiltonian NNs have been used for simulating complex physical systems (Sanchez-Gonzalez et al., 2019, 2020). The key idea of this work is to represent the states of particles as a graph and use a graph neural network (GNN) to predict the change from the current state to the next state. In a follow-up Cranmer et al. (2020), introduce sparsity on the messages in a graph and use the symbolic regression method to search for physical laws that describe the messages in the graph. Recently Toth et al. (2019) developed the Hamiltonian generative network (HGN), where they proposed to learn a Hamiltonian from image sequences. HGN maps a sequence to a latent representation and then projects it to the phase space to unroll the dynamics using a symplectic ODE integrator with Hamilton's equation. In another work Yildiz et al. (2019) use second-order ODE parameterised as a BNN for modelling dynamics of high dimensional sequence data in the latent space of VAE. Most of the developments are built on the neural ODE (Chen et al., 2018), an idea to view layers of NNs as internal states of an ODE. These methods rely on the numerical integration scheme and the stability of the ODE solver. A Hamiltonian formalism dictates an additional requirement that the dynamics of an ODE should be volume-preserving and reversible. We want to clarify that, unlike HGN, which mainly focuses on sequence generation and relies on symplectic ODE integrators in the latent space, we use linear Hamiltonian operators with matrix exponentials and demonstrate its relevance for disentanglement.

**Latent Space Models** There is a long history of latent state space models for modelling sequences (Kalman, 1960; Starner and Pentland, 1997; Roweis and Ghahramani, 1999; Elliott and Krishnamurthy, 1999; Pavlovic et al., 2000). More recently, these methods have been combined with deep generative models for generating high dimensional sequences as well as learning a disentangled representation (Karl et al., 2016; Villegas et al., 2017; Tulyakov et al., 2018; Hsieh et al., 2018; Yingzhen and Mandt, 2018; Miladinović et al., 2019; Minderer et al., 2019; Franceschi et al., 2020; Zhu et al., 2020). MoCoGAN (Tulyakov et al., 2018) developed an adversarial framework, combining a random content noise with a sequence of random motion noise to generate videos. More recently, DSVAE (Yingzhen and Mandt, 2018) proposed to split a latent space into time-variant and invariant representations and use LSTM (Hochreiter and Schmidhuber, 1997) to learn the prior on time-variant representation. S3VAE (Zhu et al., 2020) improves the disentanglement of DSVAE by minimising a mutual information loss between content and motion variables. Some Hamiltonian methods (Toth et al., 2019; Yildiz et al., 2019) also model the dynamics of high dimensional sequential data in a latent space. However, the focus in those cases is only on sequence generation; to our knowledge, this has not been investigated for disentanglement.

**Group Transformations in Latent Space Models** Rao and Ruderman (1999) proposed the algorithm to model the infinitesimal movement on data manifold using learnable Lie group operators.Culpepper and Olshausen (2009) use the matrix exponents to learn the transport operators for modelling the manifold trajectory. Many other similar methods have investigated the use of geometric operators for learning the manifold representation from data (Rao and Ruderman, 1999; Culpepper and Olshausen, 2009; Memisevic, 2012; Sohl-Dickstein et al., 2010; Cohen and Welling, 2014). The use of symmetries for learning disentangled factors of variations has recently been considered. A disentanglement is generally identified as learning representations with independent latent factors. The main goal is that each latent factor should control a distinct data factor, and a single latent variable should control no two data factors, Various authors (Bengio et al., 2013; Lake et al., 2017; Eastwood and Williams, 2018). Higgins et al. (2018) have proposed a symmetry-based definition of disentanglement. The goal in these settings was to decompose a latent space into subspaces and on each subspace, to learn a unique group transformation such that the subspace is unchanged by the action of other groups. Caselles-Dupré et al. (2019), build such a model using interaction with the environment. Some other

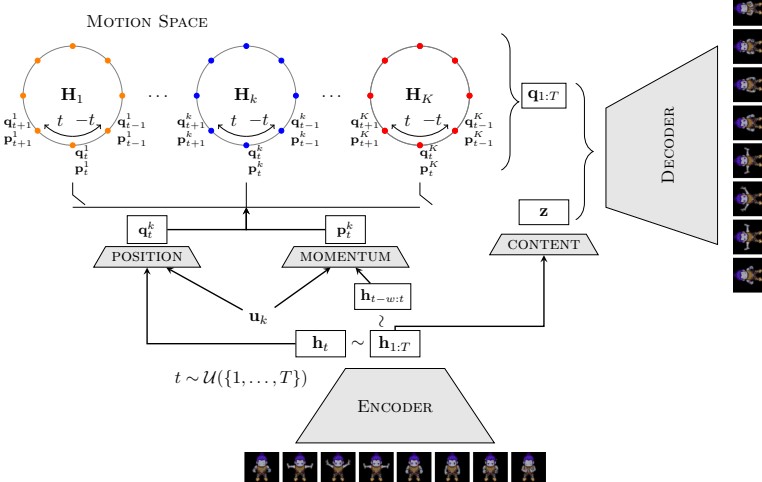

Figure 2: The framework for our model. We first encode each time step of a sequence to a respective feature vector $\mathbf{h}_{1:T}$. Next, to unroll the dynamics of an action $k$, we map the feature representation to the respective phase space. Specifically, we sample a starting index $t$ and map $\mathbf{h}_t$ to position coordinate $\mathbf{q}_t^k$. For momentum $\mathbf{p}_t^k$, we use temporal convolution with a kernel size $w$ on $\mathbf{h}_{t-w:t}$. We then use the operator $\mathbf{H}_k$ to trace out the forward and backward trajectory. At last, we combine the position coordinates of all timesteps $\mathbf{q}_{1:T}$ with the content representation $\mathbf{z}$ and pass it through the decoder network to generate the sequence.

similar approaches were recently proposed to learn group transformations in a latent space (Connor and Rozell, 2020; Quessard et al., 2020; Dupont et al., 2020). However, the applications were restricted to relatively toy problems and, to our knowledge, have not been investigated on higher dimensional videos.

## 3 Method

Here, we introduce *HALO* for sets of sequential image data. Each set of sequences depicts the temporal evolution associated with one of several *actions*. In this context, an *action* is simply a label associated with a particular sequence set, but where it is understood, the sequences within a set may have very different content but the same dynamic form, e.g. in the sprites data (discussed later) the actions are 'walking', 'spell cast', and 'slash' and the sequences within a set are different individuals performing the relevant action. In the following, we assume the separation into action sets is known, but that assumption is relaxed later.

Let $\mathbf{x}_{1:T}^i$ denote the $i$th image sequence, with $\mathbf{x}_t^i$ the $t$th frame in the sequence. Let $\mathbf{u}^i$ be an indicator vector denoting the action associated with the $i$th sequence; i.e. $u_k^i = 1$ iff sequence $i$ follows action $k$ and $u_{k'}^i = 0$ for all other $k' \neq k$. These sequences and corresponding actions are collected into a dataset $\{(\mathbf{x}_{1:T}^i, \mathbf{u}^i)\}_{i=1}^N$ of size $N$, where, for the sake of simplicity in description, we assume they all are of same length $T$. In this paper, we use a latent space to aid the modelling of each sequence and decompose that latent space into two parts, which we call a *content* space (denoted by $\mathbf{Z}$) and a *motion* space (denoted by $\mathbf{S}$). As the data comprises sequences of various actions that take different dynamical forms, we further decompose the latent motion space $\mathbf{S} = \mathbf{S}^1 \oplus \mathbf{S}^2 \oplus \ldots \oplus \mathbf{S}^K$, with one subspace for each action. In modelling a sequence corresponding to action $k$, only the subspace $\mathbf{S}^k$ will be allowed to change across the length of that sequence. Each motion subspace is further decomposed into generalised *position* and *momentum* parts: $\mathbf{S}^k = (\mathbf{p}^k, \mathbf{q}^k)$. Only the position component of this phase space is used together with content to create individual images in a sequence generatively. The momentum part *only* affects the dynamics.

The above formulation provides many advantages; it prevents the neural network from leaking constant *content* information via the motion representation, and it ensures the possibility of preserving key conservation quantities that we argue are implicit in the constraints of most motion dynamics. This is discussed further in Paragraph 3. The full framework of our model is illustrated in Figure 2.

We next introduce the generative model, followed by the variational formalism for inference and learning.

**Generative Model**  For completeness we first present the full probabilistic model in (1-4) before describing each component. Each dynamic is categorised by a particular *action* enumerated by $k$, encoded in an indicator vector $\mathbf{u}$ (i.e. $u_k = 1$ for action $k$). The generative model is conditioned on this action vector. First, in (1), we sample the *content* variable $\mathbf{z}$ from a prior $p(\mathbf{z})$. The content variable will describe the constant appearance characteristics expressed throughout the sequence. Next, we sample a starting position from a prior $p(\mathbf{q}_1^k)$, and momentum from a prior $p(\mathbf{p}_1^k)$ (we initialise the actions not represented in the sequence to zero). The full state-space representation for the dynamic of action $k$ is then given by $\mathbf{s}_1^k = (\mathbf{p}_1^k, \mathbf{q}_1^k)$. The dynamical model (3,5) then traces out the forward trajectory in the phase space. Finally, we combine the position trajectory with the content representation and use a decoder neural network to get the emission distribution of the data space sequence. In summary,

$$\text{GIVEN: } k \text{ denoting action label for a sequence,}$$

$$\mathbf{z} \sim p(\mathbf{z}), \quad \mathbf{q}_1^k \sim \mathcal{N}(\mathbf{0}, \mathbf{I}_\mathsf{d}), \quad \mathbf{p}_1^k \sim \mathcal{N}(\mathbf{0}, \mathbf{I}_\mathsf{d}), \tag{1}$$

$$\mathbf{s}_1^k = [\mathbf{p}_1^k, \mathbf{q}_1^k], \quad \mathbf{s}_1^{k'} = \mathbf{0}, \quad \forall k' \neq k, \tag{2}$$

$$\mathbf{s}_t^k = f(\mathbf{s}_{t-1}^k; \omega_k, t), \quad \mathbf{s}_t^{k'} = \mathbf{s}_{t-1}^{k'}, \quad \forall t > 1, k' \neq k \tag{3}$$

$$\mathbf{q}_t = [\mathbf{q}_t^1, \ldots, \mathbf{q}_t^K], \mathbf{x}_t \sim \mathcal{N}(\mathbf{x}_t | \phi(\mathbf{z}, \mathbf{q}_t), \alpha^2 \mathbf{I}_\mathsf{m}), \; \forall t \tag{4}$$

where $d$ is the dimensionality of $k^{th}$ subspace, $m$ is the dimensionality of data space, $f$ is a dynamical model (5) and $\omega_k$ are the parameters of $f$ to be used for the $k^{th}$ subspace. We use an emission distribution that is a spherical Gaussian, with a parameterised mean $\phi(\cdot, \cdot)$, and a covariance $\alpha^2 \mathbf{I}_\mathsf{m}$.

**Dynamical Model**  In image sequences, we can view each frame of a sequence as a point in an abstract representation space; the temporal dynamics trace a path connecting the frames forming a 1-submanifold of the image manifold. Most dynamical models either try to capture this geometry deterministically (Srivastava et al., 2015) or probabilistically (Chung et al., 2015; Hsu et al., 2017; Yingzhen and Mandt, 2018) via linear or non-linear state-space models. In either case, small errors in dynamical steps can accumulate and result in a significant deviation from the manifold when unrolling long-term trajectories at inference time (Karl et al., 2016; Fraccaro et al., 2017). Interestingly, Hamiltonian systems alleviate these issues by constraining the dynamics to be symplectic and reversible. The symplectic geometry ensures the dynamics are volume-preserving, preventing any deviation from the manifold, and reversibility is useful in understanding how the state of an object changes under dynamical evolution. By reversing the arrow of time, the object could return to its previous state, this awareness provides a sense of accountability to an object for its actions. In our work, without significant loss of generality, we propose a linear Hamiltonian system in the latent layer, relying on deep neural network mapping to data space to handle all nonlinear aspects. The linearity of dynamics also enhances the interpretability of the dynamics.

**Definition 1.** A matrix $\mathbf{H} \in \mathbb{R}^{2d \times 2d}$ is an Hamiltonian matrix if $\mathbf{H}^T \mathbf{J} \mathbf{H} = \mathbf{J}$, where $\mathbf{J}$ is a skew-symmetric matrix $\mathbf{J} = \begin{pmatrix} 0 & \mathbf{I}_\mathsf{d} \\ -\mathbf{I}_\mathsf{d} & 0 \end{pmatrix}$ and $\mathbf{I}_\mathsf{d}$ is an identity matrix.

Consider a coordinate vector $\mathbf{s} \in \mathbb{R}^{2d}$ in the phase space $\mathbf{S}$ at a time $t$ that evolves under constant Hamiltonian energy $\mathbf{E} = \frac{1}{2}\mathbf{s}^T \mathbf{M}(t)\mathbf{s}$, where $\mathbf{M}(t)$ is a symmetric matrix. In Hamiltonian mechanics, the coordinates are specified in terms of position $\mathbf{q}$ and momentum $\mathbf{p}$ variables as $\mathbf{s} = (\mathbf{q}, \mathbf{p})$. Using the fact energy $\mathbf{E}$ is constant over time we can express equation of motion as, $\frac{d\mathbf{s}(t)}{dt} = \mathbf{J}\mathbf{M}(t)\mathbf{s}$. Let $\mathbf{H}(t) = \mathbf{J}\mathbf{M}(t)$, we can rewrite the equation of motion as, $\frac{d\mathbf{s}(t)}{dt} = \mathbf{H}(t)\mathbf{s}$. The closed-form solution is given by matrix exponential $\mathbf{s}(t) = e^{t\mathbf{H}}\mathbf{s}(0)$. The matrix exponent has a connection to Lie algebras, and for small $t$ we can interpret $e^{t\mathbf{H}}$ as an infinitesimal transformation of $\mathbf{s}(0)$ under the action of a Lie group of $\mathbf{H}$. We discuss this further in Appendix 2. For a detailed introduction to the topic, we refer to Chevalley (2016). We use fast Taylor approximation (Bader et al., 2019) to compute matrix exponential that provides a stable solution under matrix norms.

In this work, we consider $K$ Hamiltonians $\mathbf{H}_1, \ldots, \mathbf{H}_K$, each acting on a unique subspace of the phase space $\mathbf{S}^1, \mathbf{S}^2, \ldots, \mathbf{S}^K$. To unroll the trajectory of motion $k$, we use the group action defined by

the matrix exponent of the operator $\mathbf{H}_k$ on a starting phase space representation $\mathbf{s}_1^k \in \mathbf{S}^k$ given by,

$$\mathbf{s}_t^k = f(\mathbf{s}_{t-1}^k; \omega_k, t) = e^{t\mathbf{H}_k}\mathbf{s}_{t-1}^k, \forall t > 1; \quad \mathbf{s}_t^{k'} = \mathbf{0}, \quad \forall t, k' \neq k. \tag{5}$$

The backward dynamics can simply be obtained by negating time, i.e. replacing $t$ with $-t$ in the above. We assume all time steps are equally spaced. The above formulation provides an explicit disentanglement of the motion space. It further allows us to parallelise the computation of matrix exponential by leveraging the block diagonal form of $\mathbf{H}$. Specific to our work, we parameterise a symmetric matrix $\mathbf{M}_k$ and obtain its Hamiltonian matrix as $\mathbf{H}_k = \mathbf{J}\mathbf{M}_k$ where $\mathbf{J}$ is a fixed skew-symmetric matrix as stated in definition (1). The group of such real Hamiltonian matrices form a symplectic Lie group under multiplication $Sp(2d)$ with $2d^2 + d$ independent elements. The symplectic geometry proves useful for long-term sequence generation. We also consider the symplectic orthogonal group $SpO(2d)$ that further restricts the Hamiltonian matrix to a skew-symmetric form with $(d^2 - d)/2$ independent elements. The benefit of this restriction is that the resulting transforms reduce to rotations which is more interpretable. We briefly introduce the details in Appendix 2. For a more comprehensive overview, we refer readers to Easton (1993).

**Inference** In order to learn the model parameters, we need to infer the distribution over latent variables. We use variational inference to learn the model parameters that leads to maximising the evidence lower bound (ELBO) objective, $\max_q \mathbb{E}_{q(\mathbf{z}, \mathbf{s}_t | \mathbf{x}_{1:T}, \mathbf{u})} \log \left[ \frac{p(\mathbf{x}_{1:T}, \mathbf{z}, \mathbf{s}_{1:T} | \mathbf{u})}{q(\mathbf{z}, \mathbf{s}_t | \mathbf{x}_{1:T}, \mathbf{u})} \right]$, where $q(.|.)$ is the approximate posterior distribution and $\mathbf{s}_t = [\mathbf{q}_t, \mathbf{p}_t]$. It remains to define the approximate posterior we use. Since the Hamiltonian dynamics are reversible, at inference time, we randomly sample a choice of frame $t$ and use forward and backward action of Hamiltonian to trace the trajectory of states after and before that frame for the respective action as stated in Equation 5.

For a sequence $\mathbf{x}_{1:T}$, we use the process in (6) to draw samples from a variational distribution $q(\mathbf{z}, \mathbf{s}_t | \mathbf{x}_{1:T}, \mathbf{u})$. Simply, we sample the content variable $\mathbf{z}$ conditioned on the observed data and independently sample the motion states $\mathbf{s}_t^k = [\mathbf{q}_t^k, \mathbf{p}_t^k]$ for the reference frame $t$ conditioned on the observed data and the relevant action $k$. Motion states corresponding to other actions are set to zero. In equations, this is,

$$\mathbf{z} \sim q(\mathbf{z}|\mathbf{x}_{1:T}), t \sim \mathcal{U}(\{1, \ldots, T\}), \mathbf{q}_t^k \sim q(\mathbf{q}_t^k|\mathbf{x}_t, \mathbf{u}), \mathbf{p}_t^k \sim q(\mathbf{p}_t^k|\mathbf{x}_{t-w:t}, \mathbf{u}), \mathbf{s}_t^k = [\mathbf{q}_t^k, \mathbf{p}_t^k] \tag{6}$$

where $t$ is a starting index, $q(\mathbf{q}_t^k|\mathbf{x}_t, \mathbf{u})$, is the posterior distributions of $k^{th}$ position subspace conditioned on the frame $\mathbf{x}_t$ and action variable $\mathbf{u}$, $q(\mathbf{p}_t^k|\mathbf{x}_{t-w:t}, \mathbf{u})$ is the posterior distributions of $k^{th}$ momentum subspace conditioned on $w$ previous frames and action variable $\mathbf{u}$ and $q(\mathbf{z}|\mathbf{x}_{1:T})$ is the posterior distribution of the content space conditioned on the entire sequence. We parameterise the factorised posterior as a spherical Gaussian distribution learned using an encoder neural network. Specifically, $q(\mathbf{z}|\mathbf{x}_{1:T})$ as a content network, $q(\mathbf{q}_t^k|\mathbf{x}_t, \mathbf{u})$ as a position network, and $q(\mathbf{p}_t^k|\mathbf{x}_{t-w:t}, \mathbf{u})$ as a momentum network. We use reparametrisation trick (Kingma and Welling, 2013) to sample from latent distribution $\mathbf{z} = \boldsymbol{\mu} + \boldsymbol{\sigma} \odot \boldsymbol{\epsilon}$ where $\boldsymbol{\epsilon} \sim \mathcal{N}(0, \mathbf{I})$.

**Learning Objective** The learning problem reduces to the optimisation of the following objective,

$$\max -KL[q(\mathbf{q}_t^k|\mathbf{x}_t, \mathbf{u})||p(\mathbf{q}_t^k)] - KL[q(\mathbf{p}_t^k|\mathbf{x}_{t-w:t}, \mathbf{u})||p(\mathbf{p}_t^k)] - KL[q(\mathbf{z}|\mathbf{x}_{1:T})||p(\mathbf{z})] + \mathbb{E}_{q(\mathbf{q}_t^k|\mathbf{x}_t, \mathbf{u})}\left[\sum_{t'} \log p(\mathbf{x}_t|\mathbf{q}_{t'}, \mathbf{z})\right].$$

We have provided the derivation of ELBO in Appendix 1. Figure 1, right side, is the probabilistic graph of the generative and inference model.

## 4 Experiments

The implementation details of neural network architecture and training procedure are discussed in Appendix 3. Our code is available on GitHub. [2] We first demonstrate the application of *HALO* on disentangling content and motion in sequences of rotating balls evolving under constant energy. Next, we investigate the applicability of our approach on two complex datasets: Sprites and MUG Aifanti et al. (2010).

**Rotating Balls** We construct a set of sequences of images of a ball that moves in an orbit under constraint $i^2 + j^2 = c$, where $(i, j)$ is a centre of a ball and $c$ is the distance from the centre of an

---

[2]`https://github.com/MdAsifKhan/HALO.git`

orbit. Each sequence is drawn from a different initial condition decided uniformly at random, and all sequences are of length 16. To introduce the content element, we colour half of the sequence as "red" and the remaining as "blue". In Figure 3, we show the result of swapping the content variable of two held-out sequences that demonstrates the effectiveness of *HALO* in disentangling *motion* from the *content* while keeping the dynamics intact.

Next, we introduce details of the two datasets, followed by a discussion of the results in Section 4.1. **Sprites** is a sequence of animated characters performing different actions ('walking', 'spell cast' and 'slashing' from three viewing angles 'left', 'right' and 'straight') as per sprites sheets.[3] The sequences are of length 8 RGB images of size $64 \times 64 \times 3$. Each character's appearance has four attributes: skin colour, hairstyle, tops and pants. Each attribute can take six values resulting in 1296 unique characters. We used 1000 characters for training and the rest for evaluation. **MUG** (Aifanti et al., 2010) a dataset of six facial expressions (anger, disgust, fear, happiness, sadness and surprise) of 52 individuals. Sequences are of variable lengths ranging from

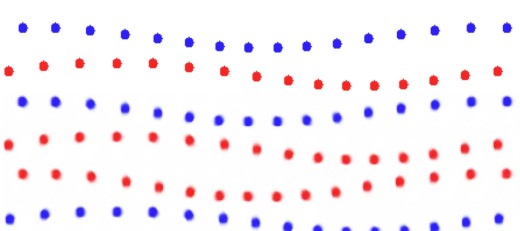

Figure 3: First two rows are original sequences, the next two rows are respective reconstructions, and the last two rows are generated by swapping the content variables. Swapping the content changes the colour but the dynamics are intact.

50 to 160 frames. We downsample the sequences by a factor of two and then take a random subsequence of length 8, crop the face region and resize it to $64 \times 64$. The training and evaluation splits are based on (Tulyakov et al., 2018). We also demonstrate the application of our model in predicting rotations on MNIST digits where the symplectic structure proves useful for sampling long trajectories in the latent space. We refer to Appendix 3.5 for results.

## 4.1 Results and Discussion

We first compare two choices of Hamiltonian structure, a symplectic group using **H** and the symplectic orthogonal group by restricting **H** to a skew-symmetric form that we refer to as skew-**H**. Here, we map a starting frame to the latent space, unroll the trajectory and then map the timesteps to a data space using a decoder. We generate sequences of length 16 (twice the length used for training purposes).

The sprites consist of periodic sequences of length 8 where the start and end frames are identical; in this case, we duplicate the sequence to get a ground truth of length 16. For MUG, we draw a sequence of length 16 from the evaluation set. We compare the generated sequences with target sequences using per-frame structural similarity index measure (SSIM), peak signal-to-noise ratio (PSNR) and mean squared error (MSE). The SSIM scores are between $-1$ and 1, with a more significant score indicating more sim-

| Model | Dataset | SSIM↑ | PSNR↑ | MSE↓ |
|---|---|---|---|---|
| **H** | Sprites | $0.982 \pm 0.005$ | $36.76 \pm 1.096$ | $0.0005 \pm 0.0002$ |
| | MUG | $0.797 \pm 0.003$ | $24.49 \pm 0.099$ | $0.0040 \pm 0.0001$ |
| Skew-**H** | **Sprites** | $0.950 \pm 0.021$ | $33.88 \pm 2.03$ | $0.0026 \pm 0.0012$ |
| | MUG | $0.791 \pm 0.003$ | $24.25 \pm 0.094$ | $0.0044 \pm 0.0001$ |

Table 1: We can see both choices of an operator can generate sequences close to the ground truth.

ilarity between the ground truth and generated sequence. Likewise, higher PSNR and lower MSE imply better generation. Table 1 describes the performance under different scores, demonstrating our model can generate high-quality sequences from an input image. We observe that **H** performs better than skew-**H**. We hypothesise that the superior performance of **H** can be attributed to the fact that skew-**H** introduces additional restrictions on the parameters of **H** that might be reducing the expressiveness of the model. This result is an interesting finding—the question we wish to investigate in future work. For the rest of the paper, we consider the dynamical operator **H**.

To evaluate disentanglement we compare with the state-of-the-art baselines DSVAE (Yingzhen and Mandt, 2018), MoCoGAN(Tulyakov et al., 2018) and S3VAE (Zhu et al., 2020).

**Quantitative Evaluation** We use a pretrained action prediction classifier for evaluating disentanglement. The architecture is provided in Table 9 of Appendix. To begin with, we draw a starting position and momentum from a prior distribution and use a dynamical model to unroll the trajectory in the phase space. Next, we sample the content variable **z** from real sequences and combine it

---

[3]https://github.com/jrconway3/ Universal-LPC-spritesheet

| Method | Data | Accuracy↑ | $H(y\|x)$↓ | $H(y)$↑ | IS↑ |
|---|---|---|---|---|---|
| *HALO* (conditional) | | **0.929** | **0.108** | **1.778** | **5.312** |
| *HALO* (unconditional) | | 0.750 | 0.187 | 1.762 | 4.830 |
| DSVAE (Yingzhen and Mandt, 2018) | MUG | 0.543 | 0.374 | 1.657 | 3.607 |
| MoCoGAN (Tulyakov et al., 2018) | | 0.631 | 0.183 | 1.721 | 4.655 |
| S3VAE (Zhu et al., 2020) | | 0.705 | 0.135 | 1.760 | 5.078 |
| *HALO* (conditional) | | **1.000** | **0.011** | 2.009 | 7.374 |
| DSVAE (Yingzhen and Mandt, 2018) | Sprites | 0.907 | 0.072 | 2.192 | 8.331 |
| MoCoGAN (Tulyakov et al., 2018) | | 0.928 | 0.090 | 2.192 | 8.182 |
| S3VAE (Zhu et al., 2020) | | **0.994** | 0.041 | **2.197** | **8.636** |

| Sprites (Attr.) | Accuracy↑ |
|---|---|
| Skin Color | 0.925 |
| Shirt | 0.948 |
| Pant | 0.968 |
| Hair | 0.992 |
| Identity (MUG) | 0.998 |

Table 2: Left shows the disentanglement of content and motion components of latent space. The high score of accuracy and Inter-Entropy $H(y)$ while low scores of Intra-Entropy $H(y|x)$ are expected from a better model. Our model performs best across all three scores on MUG. On sprites, we are comparable to S3VAE. This is due to the simplicity of classes in sprites. We want to remark our unconditional model, as demonstrated on MUG, significantly outperforms other baselines showing the benefit of Hamiltonian even when labels are not available. On the right, we investigate the extent to which content is preserved when we switch motion variables with an arbitrary sequence. We report the accuracy of individual attributes in sprites and the identity of actors in the MUG dataset. The results show the content space can capture attributes that don't change under dynamics.

with position variables to generate image sequences. We report the performance of the classifier in predicting the action from these generated sequences. The score is a useful measure of the model's tendency to keep the motion intact with the modified content. We use the same classifier to report the intra-Entropy $H(y|x)$ and inter-Entropy $H(y)$ that estimates the diversity of generated sequences. $H(y|x)$ measures the closeness of generated sequences to the real sequences, and $H(y)$ measures the diversity of generated sequences (He et al., 2018). The two scores can be combined together to obtain inception score (IS) Salimans et al. (2016) a commonly used evaluation criterion for the diversity of generated samples($IS = \exp(H(y) - H(y|x))$ Barratt and Sharma (2018)). The results are reported in Table 2. We observe *HALO* outperforms the baselines on the MUG and is comparable with S3VAE on sprites. This improvement results from explicitly associating an action with a unique subspace that allows separability of the dynamics and avoids any mixing or ambiguity of action in the motion space. The results on sprites are comparable; we attribute this to the simplicity of sprites' classes that result in high performance across all models. We want to remark that our formulation is not constrained by action variables $\mathbf{u}$. Table 2 also describes the results for an unconditional version on the MUG (Aifanti et al., 2010) that significantly outperforms the baseline. The details of the unconditional model are provided in Appendix 3.2. The benefit of incorporating action variables $\mathbf{u}$ is that it allows controlled generation of sequences, as demonstrated in Figure 6.

We, next evaluate the tendency to preserve the identity of sequences. For sprites, identity refers to four different attributes, and for MUG (Aifanti et al., 2010) it is the sequence label of the individual. We pre-train a classifier on the task of identity prediction and use it for evaluating the generated sequences. This way, we measure the model's ability to keep the identity intact when the motion is changed. For sprites, we report the accuracy of individual attributes. Table 1 outlines the results. We can see on MUG that our model can preserve the identity with high accuracy. We can make a similar observation for different attributes of sprites sequences. Thus, good performance indicates that the content is preserved when traversing the motion subspace, and the motion space is invariant when changing the content variables. Also validated by the qualitative results.

**Qualitative Evaluation** We first evaluate the quality of sequence prediction by comparing the original sequence, its reconstruction and the generation from an initial frame. To predict the future timesteps of a sequence, we apply the motion operator on the latent encoding of the first time step. Figure 4 on the left are the results for sprites, and on the right of MUG video sequences. Next, we report sequences reconstructed by swapping motion variables to evaluate disentanglement. We start by encoding two sequences $\mathbf{x}^1_{1:T}$ and $\mathbf{x}^2_{1:T}$ to their latent representations $(\mathbf{z}^1, \mathbf{q}^1_{1:T})$ and $(\mathbf{z}^2, \mathbf{q}^2_{1:T})$, next we swap the motion variables $(\mathbf{z}^1, \mathbf{q}^2_{1:T})$ and $(\mathbf{z}^2, \mathbf{q}^1_{1:T})$ between the two representation spaces, and then pass the resulting representations through the decoder to generate the sequences $\mathbf{x}^{1\rightarrow2}_{1:T}$ and $\mathbf{x}^{2\rightarrow1}_{1:T}$. Figure 5, on the left, are the pair of consecutive rows of original sequences and on the right of the sequences generated by swapping the motion representations. We can see that swapping the motion part does not affect the identity of the sequences.

We now evaluate the image-to-sequence task to investigate the suitability of our model for a controlled generation. We first encode the image to its content and phase space coordinate in different motion spaces. Next, use the respective operators to unroll the trajectories in phase space, which are combined

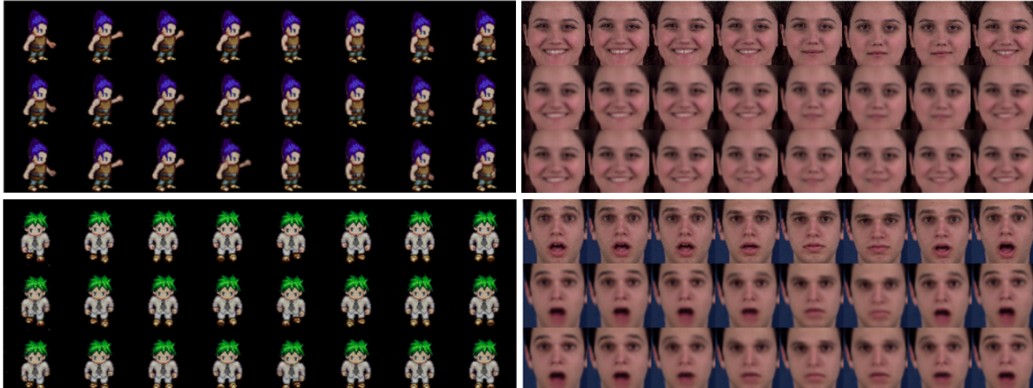

Figure 4: In each patch, the first row is the original sequence, the second row is reconstruction, and the third row is a sequence generated by an action of the operator on the starting time step. The reconstruction demonstrates our model can learn good representations, and the generation demonstrates the dynamical operator can capture realistic motions from an arbitrary starting frame. More examples are in Appendix 3.2

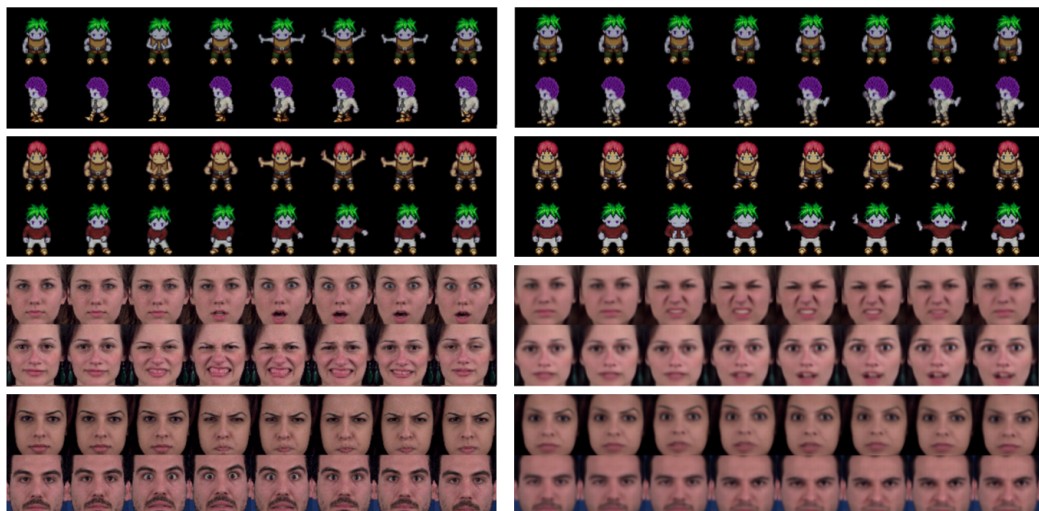

Figure 5: Qualitative demonstration of content and motion disentanglement. On the left side, we show rows of original sequence pairs and on the right are the reconstructions after swapping the motion variables in the latent space. More examples are in Appendix 3.2

with content and mapped to the image space using a decoder network. Figure 6 shows examples of decoding different motions from the same input image. For sprites, the actions are in order 'walk', 'spell card', 'slash' and for MUG they are ordered 'anger', 'disgust', 'fear', 'happiness', 'sadness' and 'surprise'. We observe that the visual dynamics associated with all the operators are well separated. We also evaluate our model for long-term sequence generation; results are presented in Appendix 3.2, where it is apparent that longer-term sequences maintain the consistency associated with the content-motion pair.

**Ablation** We carry out ablation to identify the benefits of using *HALO* over other dynamical methods. Figure 6 we compare the motion transfer by replacing Hamiltonian with a linear dynamical model and RNN. We observe that, unlike *HALO*, the variations in dynamics get constant for Linear and RNN models. This result shows the benefit of Hamiltonian dynamics as, by definition, they ensure the phase space coordinates change over time, preventing the encoder neural network from channelling any static information in the motion space, which is vital for explicit disentanglement of content and motion variables. More results are provided in Appendix 3.4. We outline the key findings in Table 3. In Figure 4 of the appendix, we show the caveat with other dynamical models on the controlled generation task.

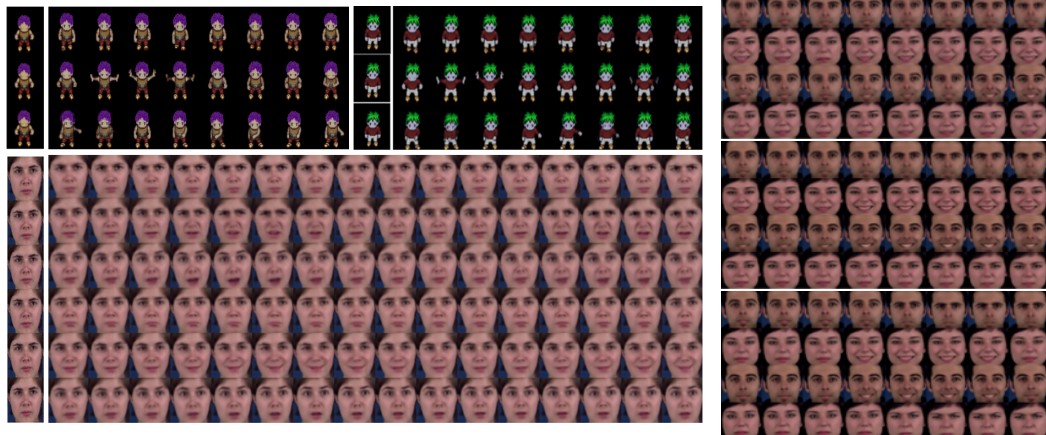

Figure 6: Left: examples of sequences generated by an action of Hamiltonian operator on the phase space representation of the starting frame. We observe that all dynamics are well separated, demonstrating the disentanglement of different actions. This result shows the benefit of the symplectic structure in motion space. Right: we demonstrate that our method outperforms other dynamical model baselines used for disentanglement. In each patch, rows one and two are original and rows three and four are obtained by swapping the motion components of latent space. The top is a Linear Model, the centre is RNN, and the last is *HALO*.

Our formulation proves useful for learning disentangled representations outperforming various baselines. We demonstrated its use for the controlled generation of image sequences. The effectiveness of our approach is a direct consequence of the *symplectic* geometry in the phase space, which prevents trajectories from deviating from the motion manifold. The choice of the quadratic form of energy provides a relationship among latent components, which we speculate is critical for the interpretability of actions.

| Dynamics | Image-To-Seq. | Motion Swap | Structure |
|---|---|---|---|
| *HALO* | ✓ | ✓ | Symplectic |
| Linear | ✗ | ✓ | ✗ |
| RNN | ✗ | ✓ | ✗ |
| Positional Encoding | ✗ | ✗ | ✗ |

Table 3: The benefits of our formulation over other dynamical models. The details on positional encoding are provided in Appendix 3.3.

## 5 Conclusions and Future Work

We introduced *HALO* – a DGM to disentangle motion from the content in image sequences. Our formulation utilises Hamiltonian latent operators to associate conserved quantities with the dynamics. Moreover, by the definition of the Hamiltonian dynamics, the motion space has to vary over time; this prevents the encoder neural network from channelling static information in the motion variables and therefore provides a helpful notion of disentanglement of content and motion. Our quantitative results with both conditional and unconditional models outperform the existing baselines. We furthermore demonstrate disentanglement qualitatively using motion swapping. In the conditional model, we associate every action with a unique Hamiltonian that proves critical for the controlled generation task of an image-to-sequence generation. *HALO* can generate long-term trajectories and traverse the motion manifolds of different actions in the latent space. We look forward to future applications to other sequential data types, such as molecular trajectories. A potential limitation of our model is that it is less able to deal with irregularly sampled sequences, changes in tempo or reversals. In future work, we wish to address this issue by allowing a more flexible prior on the spacing between time steps.

**Societal Impact** The applications of generative models for the realistic generation of images or videos is an immediate concern for society. The proposed application, like motion swap and controlled generation of videos, can be potentially misused for generating fake data. We restrict our paper's scope to research and use data within the condition in the license agreement.

### Acknowledgements

The authors would like to thank Joseph Mellor and William Toner for the helpful discussion during the project and Elliot J. Crowley for the valuable feedback on the paper. This research was funded in part by an unconditional gift from Huawei Noah's Ark Lab, London.

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
