# Supplementary Material

## 1 Derivation of ELBO

We use maximum loglikelihood on sequence variables to derive the evidence lower bound (ELBO),

$$
\begin{aligned}
\log p(\mathbf{x}_{1:T}|\mathbf{u}) &= \log \int p(\mathbf{x}_{1:T}, \mathbf{z}, \mathbf{s}_{1:T}|\mathbf{u}) d\mathbf{s}_{1:T} d\mathbf{z} \\
&= \log \int \frac{p(\mathbf{x}_{1:T}, \mathbf{z}, \mathbf{s}_{1:T}|\mathbf{u})}{q(\mathbf{z}, \mathbf{s}_t|\mathbf{x}_{1:T}, \mathbf{u})} q(\mathbf{z}, \mathbf{s}_t|\mathbf{x}_{1:T}, \mathbf{u}) d\mathbf{s}_{1:T} d\mathbf{z} \\
&\geq \int \log \left[ \frac{p(\mathbf{x}_{1:T}, \mathbf{z}, \mathbf{s}_{1:T}|\mathbf{u})}{q(\mathbf{z}, \mathbf{s}_t|\mathbf{x}_{1:T}, \mathbf{u})} \right] q(\mathbf{z}, \mathbf{s}_t|\mathbf{x}_{1:T}, \mathbf{u}) d\mathbf{s}_{1:T} d\mathbf{z} \\
&\geq \mathbb{E}_{q(\mathbf{z}, \mathbf{s}_t|\mathbf{x}_{1:T}, \mathbf{u})} \log \left[ \frac{p(\mathbf{x}_{1:T}, \mathbf{z}, \mathbf{s}_{1:T}|\mathbf{u})}{q(\mathbf{z}, \mathbf{s}_t|\mathbf{x}_{1:T}, \mathbf{u})} \right]
\end{aligned}
\tag{1}
$$

where $\mathbf{s}_t = [\mathbf{q}_t, \mathbf{p}_t]$. The joint distribution is factorised as,

$$
p(\mathbf{x}_{1:T}, \mathbf{z}, \mathbf{s}_{1:T}|\mathbf{u}) = p(\mathbf{z})p(\mathbf{x}_1|\mathbf{q}_1, \mathbf{z}) \prod_{t=1}^{T-1} p(\mathbf{x}_{t+1}|\mathbf{q}_{t+1}, \mathbf{z})p(\mathbf{q}_{t+1}, \mathbf{p}_{t+1}|\mathbf{q}_t, \mathbf{p}_t, \mathbf{u})
\tag{2}
$$

Since, we transform the starting latent state $\mathbf{s}_1 = [\mathbf{q}_1, \mathbf{p}_1]$ using a deterministic transformation $f(t, \mathbf{H}; \omega) = e^{t\mathbf{H}}$ (where $\omega$ are the parameters of $\mathbf{H}$ matrix), we can write our transition distribution as,

$$
p(\mathbf{s}_{t+1}|\mathbf{s}_t, \mathbf{u}) = p(\mathbf{s}_t|\mathbf{s}_{t-1}, \mathbf{u}) \left| \frac{df}{d\mathbf{s}_t} \right| = p(\mathbf{s}_t|\mathbf{s}_{t-1}, \mathbf{u})e^{\mathrm{Tr}(\mathbf{H})} = p(\mathbf{s}_1|\mathbf{u}) \prod^{t} e^{\mathrm{Tr}(\mathbf{H})}
\tag{3}
$$

where Tr is the trace operator and $p(\mathbf{s}_1|\mathbf{u}) = p(\mathbf{q}_1|\mathbf{u})p(\mathbf{p}_1|\mathbf{u})$. The transition model is reversible; therefore, without loss of generality, we can replace a starting step 1 with any arbitrary $t$ and unroll both forward and backwards. We next equate (3) in the generative model defined in (2) that reduces the factorisation to,

$$
p(\mathbf{x}_{1:T}, \mathbf{z}, \mathbf{s}_{1:T}|\mathbf{u}) = p(\mathbf{z})p(\mathbf{x}_1|\mathbf{q}_1, \mathbf{z})p(\mathbf{q}_t|\mathbf{u})p(\mathbf{p}_t|\mathbf{u}) \prod_{t'=1, \neq t}^{T-1} p(\mathbf{x}_{t'}|\mathbf{q}_{t'})e^{\mathrm{Tr}(\mathbf{H})}
\tag{4}
$$

We factorise the variational distribution $q(\mathbf{z}, \mathbf{s}_t|\mathbf{x}_{1:T}, \mathbf{u})$ as,

$$
q(\mathbf{z}, \mathbf{s}_t|\mathbf{x}_{1:T}, \mathbf{u}) = q(\mathbf{z}|\mathbf{x}_{1:T})q(\mathbf{q}_t|\mathbf{x}_t, \mathbf{u})q(\mathbf{p}_t|\mathbf{x}_{t-w:t}, \mathbf{u}), \quad \mathbf{s}_t = [\mathbf{q}_t, \mathbf{p}_t]
\tag{5}
$$

We now use the equations (5) and (4) to rewrite the ELBO as,

$$
\mathbb{E}_{q(\mathbf{z}|\mathbf{x}_{1:T}), q(\mathbf{q}_t|\mathbf{x}_t, \mathbf{u}), q(\mathbf{p}_t|\mathbf{x}_{t-w:t}, \mathbf{u})} \log \left[ \frac{p(\mathbf{z})p(\mathbf{q}_t|\mathbf{u})p(\mathbf{p}_t|\mathbf{u})p(\mathbf{x}_1|\mathbf{q}_1, \mathbf{z}) \prod_{t'=1, \neq t}^{T} p(\mathbf{x}_{t'}|\mathbf{q}_{t'}, \mathbf{z})e^{Tr(\mathbf{H})}}{q(\mathbf{z}|\mathbf{x}_{1:T})q(\mathbf{q}_t|\mathbf{x}_t, \mathbf{u})q(\mathbf{p}_t|\mathbf{x}_{t-w:t}, \mathbf{u})} \right]
\tag{6}
$$

$$
\mathbb{E}_{q(\mathbf{q}_t|\mathbf{x}_t, \mathbf{u})} \log \left[ \frac{p(\mathbf{q}_t|\mathbf{u})}{q(\mathbf{q}_t|\mathbf{x}_t, \mathbf{u})} \right] + \mathbb{E}_{q(\mathbf{p}_t|\mathbf{x}_{t-w:t}, \mathbf{u})} \log \left[ \frac{p(\mathbf{p}_t|\mathbf{u})}{q(\mathbf{p}_t|\mathbf{x}_{t-w:t}, \mathbf{u})} \right] + \mathbb{E}_{q(\mathbf{z}|\mathbf{x}_{1:T})} \log \left[ \frac{p(\mathbf{z})}{q(\mathbf{z}|\mathbf{x}_{1:T})} \right]
$$

$$
+ \mathbb{E}_{q(\mathbf{q}_t|\mathbf{x}_t, \mathbf{u})} \left[ \sum_{t'} \log p(\mathbf{x}_{t'}|\mathbf{q}_{t'}, \mathbf{z}) \right]
\tag{7}
$$

The trace of the real-Hamiltonian matrix is zero we can therefore omit the term $Tr(\mathbf{H})$. Since, for each motion $\mathbf{u}_k$ we associate a separate Hamiltonian $\mathbf{H}_k$ that acts on a subspace $\mathbf{S}^k$, we can view the full state space $\mathbf{S}$ as a partitions of symmetry groups $\mathbf{S} = \mathbf{S}_1 \oplus \cdots \oplus \mathbf{S}_K$ where the Hamiltonian $\mathbf{H}$ is in the block diagonal form $\mathbf{H} = diag(\mathbf{H}_1, \cdots, \mathbf{H}_K)$. We, therefore, express the distributions in terms of the variables of their respective subspaces to obtain the final ELBO,

$$-KL[q(\mathbf{q}_t^k|\mathbf{x}_t, \mathbf{u})||p(\mathbf{q}_t^k)] - KL[q(\mathbf{p}_t^k|\mathbf{x}_{t-w:t}, \mathbf{u})||p(\mathbf{p}_t^k)] - KL[q(\mathbf{z}|\mathbf{x}_{1:T}, \mathbf{u})||p(\mathbf{z})]$$

$$+ \mathbb{E}_{q(\mathbf{q}_t^k|\mathbf{x}_t, \mathbf{u})} \left[ \sum_{t'} \log p(\mathbf{x}_{t'}|\mathbf{q}_{t'}, \mathbf{z}) \right] \tag{8}$$

## 2  Background

In this section, we provide a short overview of the definitions relevant to the context of our work. The symmetry of an object is a transformation that leaves some of its properties unchanged. E.g., translation, rotation, etc. The study of symmetries plays a fundamental role in discovering the constants of physical systems. For instance, space translation symmetry means the conservation of linear momentum, and rotation symmetry implies the conservation of angular momentum. Groups are fundamental tools used for studying symmetry transformations. Formally we say,

**Definition 1.** A group $G$ is a set with a binary operation $*$ satisfying the following conditions:

- closure under $*$, i.e., $x * y \in G$ for all $x, y \in G$

- there is an identity element $e \in G$, satisfying $x * e = e * x = e$ for all $x \in G$

- for each element $x \in G$ there exist an inverse $x^{-1} \in G$ such that $x * x^{-1} = x^{-1} * x = e$

- for all $x, y, z \in G$ the associative law holds i.e. $x * (y * z) = (x * y) * z$

The nature of the symmetry present in a system decides whether a group is discrete or continuous. A group is discrete if it has a finite number of elements. For e.g., a dihedral group $D2$ generated using an $e$ identity, $r$ rotation by $\pi$, and $f$ reflection along x-axis consists of finite elements $\{e, r, f, rf\}$. The group generators are a set of elements that can generate other group elements using the group multiplication rule. For $D2$ the generators are $\{e, r, f\}$. A continuous group is characterised by the notion of infinitesimal transformation and is generally known as the Lie group.

**Definition 2.** A Lie group $G$ is a group which also forms a smooth manifold structure, where the group operations under multiplication $G \times G \to G$ and its inverse $G \to G$ are smooth maps.

A group of 2D rotations in a plane is one common example of Lie group given by, $\mathbf{SO}(2) = \{R \in \mathbb{R}^{2 \times 2} | R^T R = I, det(R) = 1\}$. $\mathbf{SO}(2)$ is a single parameter group simply given by a 2D rotation matrix $R(\theta) = \begin{pmatrix} \cos\theta & -\sin\theta \\ \sin\theta & \cos\theta \end{pmatrix}$.

**Definition 3.** A Lie algebra $\mathfrak{g}$ of a Lie group $G$ is the tangent space to a group defined at its identity element $I$ with an exponential map $exp : \mathfrak{g} \to G$ and a binary operation $\mathfrak{g} \times \mathfrak{g} \to \mathfrak{g}$.

The structure of Lie groups is of much interest due to Noether's theorem, which states that a conservation law exists for any differentiable symmetry. In physics, such conservation laws are studied by identifying the Hamiltonian of the physical system (Easton, 1993). In this work, we look at two choices of Hamiltonians that form a symplectic group $Sp(2d)$ and symplectic orthogonal group $SpO(2d)$ structure.

**Definition 4.** A symplectic group $Sp(2d)$ is a Lie group formed by the set of real symplectic matrices defined as $Sp(2d) = \{\mathbf{H} \in \mathbb{R}^{2d \times 2d}| \quad \mathbf{H}^T \mathbf{J} \mathbf{H} = \mathbf{J}\}$, where $\mathbf{J} = \begin{pmatrix} 0 & \mathbf{I_d} \\ -\mathbf{I_d} & 0 \end{pmatrix}$.

**Definition 5.** The Lie algebra $\mathfrak{sp}$ of a symplectic group $Sp(2d)$ is a vector space defined by, $\mathfrak{sp} = \{\mathbf{H} \in \mathbb{R}^{2d \times 2d}| \quad \mathbf{J}\mathbf{H} = (\mathbf{J}\mathbf{H})^T\}$

**Definition 6.** A symplectic orthogonal group $SpO(2d)$ is defined by restricting the Hamiltonian matrices to be of orthogonal form.

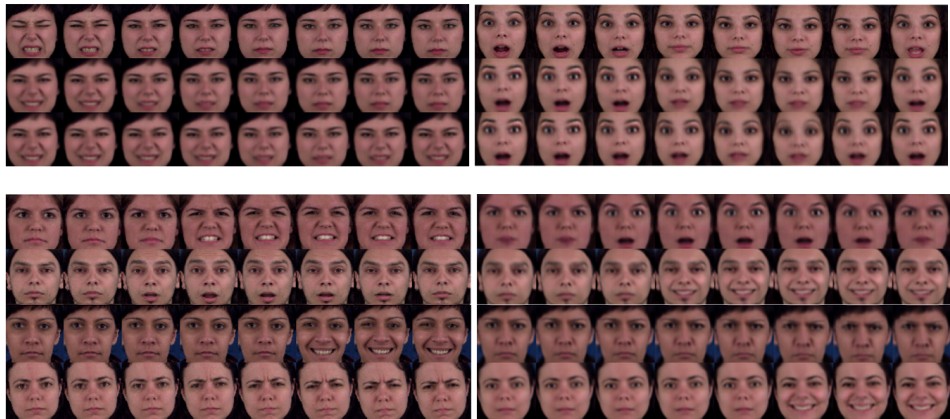

Figure 1: Unconditional Hamiltonian approach. On top, the first row is an original sequence, the second row is the reconstruction, and the third row is generated by an action of Hamiltonian on the phase space representation of the first frame in the sequence. On the bottom is an example of a motion swap, on the left side are two original motions and on the right side are sequences generated by swapping the motion variables.

**Definition 7.** A group action is a map $\circ : G \times \mathcal{X} \to \mathcal{X}$ iff (i) $e \circ x = x, \forall x \in \mathcal{X}$, where $e$ is the identity element of $G$, (ii) $(g_1 \cdot g_2) \circ x = g_1 \cdot (g_2 \circ x), g_1, g_2 \in G, \forall x \in \mathcal{X}$ where $\cdot$ is a group operation.

## 3   Experiment and Results

### 3.1   Network Architecture

The architecture of the encoder and decoder network is based on (Yingzhen and Mandt, 2018) also outlined in Table 3 and 4. We use the same network architecture for both sprites and the MUG dataset. The output of an encoder is fed to the content, position, and momentum network to get the variational distributions in **Z**, **Q** and **P** space. Table 5 describes the architecture of the network. For the position and momentum network, the input action $k$ is represented by a one-hot vector **u** that takes one at index $k$ and is zero elsewhere.

#### 3.1.1   Training details

For MUG, we choose $|\mathbf{Z}| = 512$, $|\mathbf{Q}| = K \times 12$ and $|\mathbf{P}| = K \times 12$ and for sprites $|\mathbf{Z}| = 256$, $|\mathbf{Q}| = K \times 6$ and $|\mathbf{P}| = K \times 6$, where $K$ is the number of actions. For sprites, $K = 3$ and for MUG $K = 6$. To train all our models, we use an Adam (Kingma and Ba, 2014) optimiser with a learning rate of $2e^{-4}$ and a batch size of $24$. We use Pytorch (Paszke et al., 2019) for the implementation. The code will be made available on publication. We train all our models on Nvidia GeForce RTX 2080 GPUs.

### 3.2   Results and Discussion

We further provide extended qualitative samples of our model on the MUG and sprites dataset. Figure (6) shows results of conditional sequence generation, Figure (9) shows results of motion swapping. Figure (7, 8) further shows examples of image to sequence generation. We generate 16 frames in future conditioned on an initial starting frame. Next, we adapt our model to scenarios where action variables are **u** not available.

**Unconditional Dynamics** In our formulation introduced in Section **??**, we use the action variable **u** to map the sequence to its respective phase space that allows the separability of dynamics and controlled generation of motion sequences. The choice to use action variables do not restrict the Hamiltonian dynamics; in this section, we adapt our formulation to sequences where action variables are not available. Specifically, we factorise the phase space into $K$ symmetry groups where the Hamiltonian

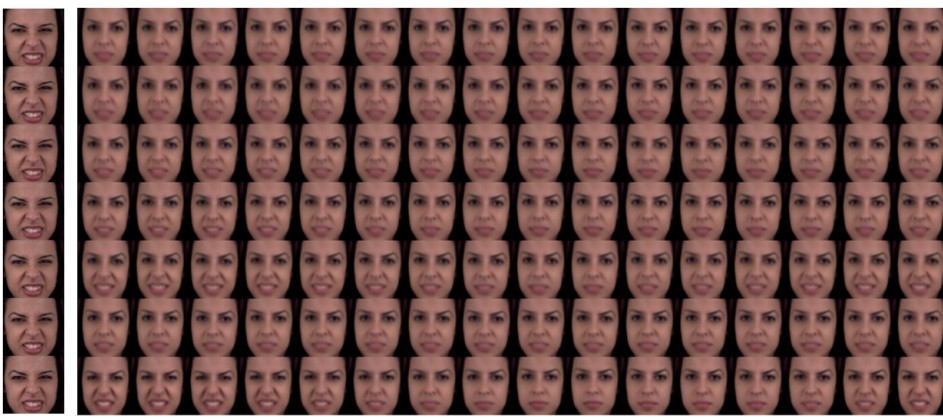

Figure 2: Unconditional Hamiltonian approach. An example of image-to-sequence generation. The first column is the starting frame, the first six rows correspond to the sequence generated by the action of $k-th$ block of $\mathbf{H}$, and the last row is the sequence generated by the full $\mathbf{H}$.

takes the form $\mathbf{H} = block\text{-}diagonal(\mathbf{H}_1, \ldots, \mathbf{H}_K)$. To unroll the trajectory for any arbitrary sequence $\mathbf{x}_{1:T}$ we evolve all the operators simultaneously as,

$$\mathbf{s}_t = f(\mathbf{s}_{t-1}^k; \omega_k, t) = block\text{-}diagonal(e^{t\mathbf{H}_1}\mathbf{s}_{t-1}^1, \ldots, e^{t\mathbf{H}_K}\mathbf{s}_{t-1}^K) \quad \forall t > 1 \tag{9}$$

We want to remark that in such a formulation, we don't have direct control over the action generated by dynamics. The type of motion generated depends on the initial position and momenta variable. Furthermore, the operators $\mathbf{H}_k$ may not necessarily correspond to specific action but could describe a more general property that is conserved and shared across motions. For instance, different operators could capture the varying magnitude of action movements like smiling, surprise, etc. To investigate it empirically, we map a starting frame to phase space and generate a sequence using individual $\mathbf{H}_k$ as well as $\mathbf{H}$. Figure 2 describes the generated motion sequences. The first six rows are sequences generated by individual $\mathbf{H}_k$, and the combined $\mathbf{H}$ generates the last row. We can observe the operators capture the varying extent of motion. Figure 1 further shows the performance of a model on sequence generation and motion transfer.

### 3.3 Ablation

To investigate the effectiveness of our dynamical model, we perform the following ablation studies,

**What is the benefit of Constant Energy?** The Hamiltonian formulation maintains the constant energy over time. Such a choice is beneficial for generating long-term sequences. In this part, we generate long sequences using our dynamical model and look at the evaluation of energy over time.

The total Hamiltonian energy in the phase space is given by,

$$\mathbf{E} = \frac{1}{2}\mathbf{s}^T\mathbf{M}\mathbf{s} \tag{10}$$

where $\mathbf{s} = (\mathbf{q}, \mathbf{p})$, and $\mathbf{M}$ is a symmetric matrix. Let $\mathbf{M}$ be a $2 \times 2$ block matrix $\mathbf{M} = \begin{pmatrix} \mathbf{A} & \mathbf{B} \\ \mathbf{B} & \mathbf{C} \end{pmatrix}$. We can expand the energy term as,

$$\mathbf{E} = \frac{1}{2}\mathbf{q}^T\mathbf{A}\mathbf{q} + \frac{1}{2}\mathbf{p}^T\mathbf{C}\mathbf{p} + \frac{1}{2}\mathbf{q}^T\mathbf{A}\mathbf{p} + \frac{1}{2}\mathbf{p}^T\mathbf{B}\mathbf{q} \tag{11}$$

The first term is potential energy (PE), the second is kinetic energy (KE), and the last two combined are non-separable terms. When B is zero, the energy is entirely separable into KE and PE terms. The non-separable Hamiltonian is common in many physical problems, for instance, rigid body dynamics

and many others appearing in quantum mechanics. For details, we refer to Easton (1993). The choice of the unconstrained linear form of Hamiltonian was motivated to allow more flexibility to the model to learn in a data-driven way.

In Figure 3, we report the plot of energy over time for an image under different motion dynamics. The change in the individual energy shows the dynamics are not constant; this is also evident from the corresponding image sequences shown in the plot. As dynamics evolve, the total energy is strictly conserved, demonstrating that the trajectory cannot diverge from the learned symplectic structure. The results demonstrate the benefit of our model in generating long-term sequences. We want to add a remark that the energy terms should be interpreted with care. It might not have any equivalence to the energy of a physical system; what it does is that it provides constraints to use the time translation symmetry of the dynamics.

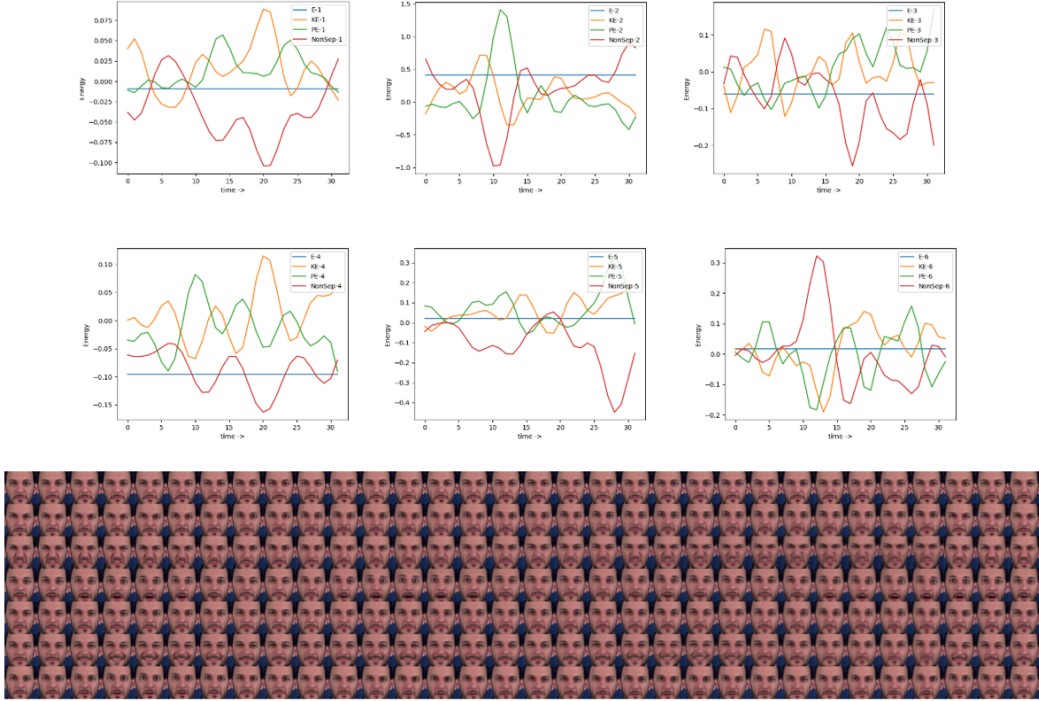

Figure 3: We map a starting frame to the phase space and use the operators $\mathbf{H}$ to generate the phase space trajectory, which is then mapped to data space using the decoder network. At the top is the plot of energy vs time of the operators $\mathbf{H}_k$ ($E$ is the total energy, $KE$ is the kinetic energy term, $PE$ is the potential energy, and $NonSep$ is the non-separable term). Below, each row is the sequence generated by the action of $\mathbf{H}_k$.

**Linear Model** A linear dynamical is defined as,

$$\mathbf{h}_t = \mathbf{A}_{t-1}\mathbf{s}_{t-1} + \mathbf{B}_{t-1}\mathbf{h}_{t-1} + \mathbf{b} \tag{12}$$

where $\mathbf{h}_t$ is a hidden state, and $\{\mathbf{A}, \mathbf{B}, \mathbf{b}\}$ are learnable parameters. To generate the trajectory $\mathbf{x}_{1:T}$, we combine the state coordinates $\mathbf{h}_{1:T} = \{\mathbf{h}_1, \ldots, \mathbf{h}_T\}$ with the content variable $\mathbf{z}$ and pass the joint representation through the decoder network. We report the performance of a linear model in conditional as well as unconditional settings.

**Positional Encoding** We generate a simplistic baseline using a fixed Fourier encoding representation. Specifically, for a sequence of frames $\mathbf{x}_{1:T} = \{\mathbf{x}_1, \ldots, \mathbf{x}_T\}$ we map it to a content variable $\mathbf{z}$ and a frame $\mathbf{x}_t$ to a phase $\phi_t \in [-1, 1]$. We then generate $T - t$ linearly separated phase coordinates $\{\phi_t, \ldots, \phi_T\} \in [\phi_t, 1 + \phi_t]$ and define the motion space representation as,

$$\mathbf{s}_t = \{\sin(\phi_t 2^k), \cos(\phi_t 2^k)\}_{k=1}^{k=\lfloor d/2 \rfloor} \tag{13}$$

where $d$ is the size of motion space. We impose a Gaussian prior on the phase coordinates $p(\phi_t) = \mathcal{N}(0, 1)$.

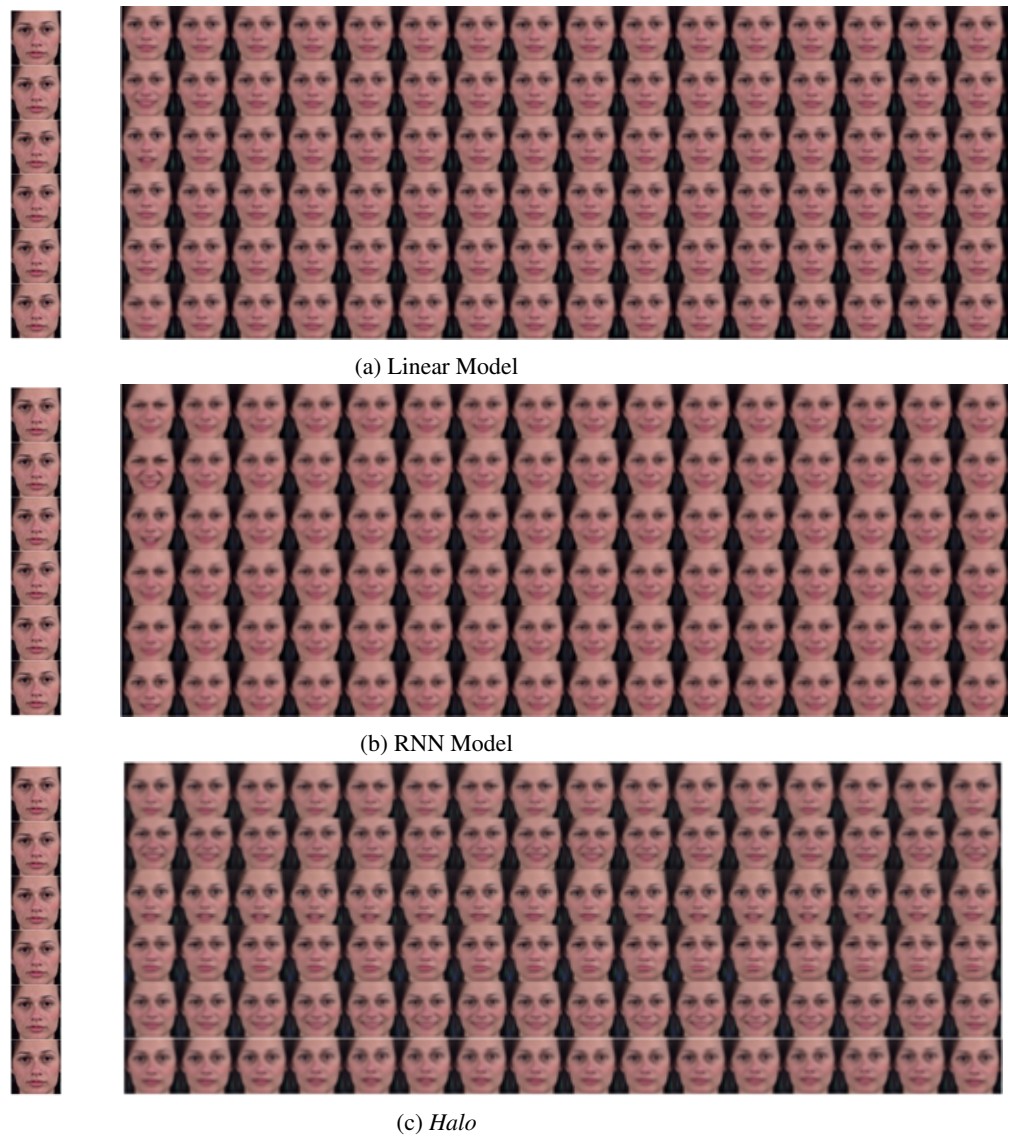

(a) Linear Model

(b) RNN Model

(c) *Halo*

Figure 4: Results on Image to sequence generation. On the left is the starting frame and on the right are different motions generated by the dynamical models.

## 3.4 Discussion

In this section, we compare *Halo* with other choices of dynamical models. We describe the qualitative results in Table 1.The Hamiltonian model achieves the best performance across all scores. We observe all models except the position encoding achieve comparable performance on identity prediction. We speculate this could be due to the non-changing dynamics, which makes predicting the identity from a sequence of static images much easier for a classifier. Due to the failure of positional encoding, we omit it from the rest of the discussion. We restrict the qualitative analysis to conditional models. Figure 4 describes the results on image-to-sequence generation, further demonstrating that the Hamiltonian dynamics are consistent in long-term prediction and prevent the flow of constant information to motion variables. Overall the Hamiltonian formulation outperforms other approaches and works best across all tasks.

| Method | Accuracy↑ | $H(y\|x)$↓ | $H(y)$ ↑ | IS ↑ |
|---|---|---|---|---|
| *HALO* | **0.929** | **0.108** | **1.778** | **5.312** |
| Linear | 0.548 | 0.722 | 1.553 | 2.295 |
| RNN | 0.580 | 0.759 | 1.743 | 2.675 |
| *HALO* (unconditional) | 0.750 | 0.187 | 1.762 | 4.830 |
| Linear (unconditional) | 0.451 | 0.962 | 1.525 | 1.756 |
| RNN (unconditional) | 0.550 | 1.015 | 1.658 | 1.902 |
| Positional Encoding | 0.152 | 0.978 | 1.150 | 1.188 |

(a) Results of a classifier on MUG for different choices of dynamical models. The high score of accuracy and Inter-Entropy $H(y)$ while low scores of Intra-Entropy $H(y|x)$ are expected from a better model.

| Identity | Accuracy↑ |
|---|---|
| *HALO* | 0.998 |
| Linear | 0.996 |
| RNN | **1.000** |
| *HALO* (unconditional) | 0.994 |
| Linear (unconditional) | 0.974 |
| RNN (unconditional) | 0.998 |
| Positional Encoding | 0.009 |

(b) Comparison to other baselines in terms of accuracy of the identity of sequences. This shows our model can preserve content when the motion representation is changed.

Table 1: Quantitative evaluation of disentanglement and diversity of generated samples

| Model | MSE ↓ |
|---|---|
| GPPVAE-dis (Casale et al., 2018) | 0.0306 |
| GPPVAE-joint (Casale et al., 2018) | 0.0280 |
| ODE$^2$VAE (Yildiz et al., 2019) | 0.0204 |
| ODE$^2$VAE-KL (Yildiz et al., 2019) | 0.0184 |
| *Halo* (Ours) | 0.0208 |

Table 2: Mean squared error on test set of rotating MNIST.

## 3.5 Rotating MNIST

In this section, we investigate the performance of our approach in predicting the rotations of MNIST digits. We use an unconditional version of our model for this part. Following the procedure of Casale et al. (2018), we generated sequences of 16 time steps by rotating the images of digit "3". We followed the same training procedure. In Figure 5, part (a) first row is the input sequence, and the second row is a reconstruction. In part (b), we show three sequences generated by random initial phase space coordinates. The network architecture for MNIST experiments is outlined in the Table (6, 7, 8).

Next, in Table 2, we compare the mean squared error (MSE) of our model with the other related methods (Yildiz et al., 2019; Casale et al., 2018). Our model achieves comparable performance to GPPVAE. The ODE$^2$VAE performs best in terms of MSE; this can be attributed to using a second-order latent ODE model. In contrast, our formulation only uses first-order dynamics, which provides extra computational efficiency. Furthermore, compared to GPPVAE, we don't have costly kernel computations.

We want to remark datasets such as stochastic movingMNIST Denton and Birodkar (2017) used in a few disentanglement papers is not a good application of our model. This is due to the nature of dynamics generated by an action of a random transformation. The Hamiltonian model relies on a dataset of data sequences where dynamics follow a conserved quantity and can be associated with constant energy. This assumption may or may not hold for SMNIST data due to random movements.

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

| Encoder Architecture of Sprites and MUG | |
| --- | --- |
| Conv2d | kernels: 256, kernelSize: $(5,5)$, stride: $(1,1)$, padding: $(2,2)$ |
| | BatchNorm2d $\rightarrow$ LeakyReLU(0.2) |
| Conv2d | kernels: 256, kernelSize: $(5,5)$, stride: $(2,2)$, padding: $(2,2)$ |
| | BatchNorm2d $\rightarrow$ LeakyReLU(0.2) |
| Conv2d | kernels: 256, kernelSize: $(5,5)$, stride: $(2,2)$, padding: $(2,2)$ |
| | BatchNorm2d $\rightarrow$ LeakyReLU(0.2) |
| Conv2d | kernels: 256, kernelSize: $(5,5)$, stride: $(2,2)$, padding: $(2,2)$ |
| | BatchNorm2d $\rightarrow$ LeakyReLU(0.2) |
| Conv2d | kernels: 256, kernelSize: $(5,5)$, stride: $(1,1)$, padding: $(2,2)$ |
| BatchNorm2d $\rightarrow$ LeakyReLU(0.2) $\rightarrow$ Rearrange('b c w h -> b (c w h)') | |
| Linear | in:=(c w h), out: 4096 |
| | BatchNorm1d $\rightarrow$ LeakyReLU(0.2) |
| Linear | in: 4096, out: 2048 |
| | BatchNorm1d $\rightarrow$ LeakyReLU(0.2) |
| Linear | in: 2048, out: $h$ |
| | BatchNorm1d $\rightarrow$ LeakyReLU(0.2) |

Table 3: Encoder network

| Decoder Architecture of Sprites and MUG | |
| --- | --- |
| Linear | in: $h$, out: 4096 |
| | BatchNorm1d $\rightarrow$ LeakyReLU(0.2) |
| Linear | in: 4096, out:(c w h) |
| BatchNorm1d $\rightarrow$ LeakyReLU(0.2) $\rightarrow$ Rearrange('b (c w h) -> b c w h') | |
| ConvTranspose2d | kernels: 256, kernelSize: $(5,5)$, stride: $(2,2)$, padding: $(2,2)$ |
| | BatchNorm2d $\rightarrow$ LeakyReLU(0.2) |
| ConvTranspose2d | kernels: 256, kernelSize: $(5,5)$, stride: $(2,2)$, padding: $(2,2)$ |
| | BatchNorm2d $\rightarrow$ LeakyReLU(0.2) |
| ConvTranspose2d | kernels: 256, kernelSize: $(5,5)$, stride: $(2,2)$, padding: $(2,2)$ |
| | BatchNorm2d $\rightarrow$ LeakyReLU(0.2) |
| ConvTranspose2d | kernels: 256, kernelSize: $(5,5)$, stride: $(2,2)$, padding: $(2,2)$ |
| | BatchNorm2d $\rightarrow$ LeakyReLU(0.2) |
| ConvTranspose2d | kernels: 256, kernelSize: $(5,5)$, stride: $(1,1)$, padding: $(2,2)$ |
| | BatchNorm2d $\rightarrow$ Tanh() |

Table 4: Decoder network

| Content and Motion Architecture of Sprites and MUG | | | | | |
| --- | --- | --- | --- | --- | --- |
| Content | | Position | | Momentum | |
| LSTM | in: $h$, out: $z$ | Linear | in: $h+k$, out: $v$ | Linear | in: $h+k$, out: $v$ |
| Linear$_\mu$ | in: $z$, out: $z$ | BatchNorm1d $\rightarrow$ LeakyReLU(0.2) | | BatchNorm1d $\rightarrow$ LeakyReLU(0.2) | |
| Linear$_{\log \sigma}$ | in: $z$, out: $z$ | Linear | in: $v$, out: $v$ | Linear | in: $v$, out: $v$ |
| | | BatchNorm1d $\rightarrow$ LeakyReLU(0.2) | | BatchNorm1d $\rightarrow$ LeakyReLU(0.2) | |
| | | Linear$_\mu$ | in: $v$, out: $q$ | TCN | kernelSize: 3, pad: 2, stride: 1 |
| | | Linear$_{\log \sigma}$ | in: $v$, out: $q$ | Linear$_\mu$ | in: $v$, out: $p$ |
| | | | | Linear$_{\log \sigma}$ | in: $v$, out: $p$ |

Table 5: Content and Motion network. TCN stands for temporal convolution network.

| Encoder Architecture MNIST | |
| --- | --- |
| Conv2d | kernels: 32, kernelSize: $(5,5)$, stride: $(2,2)$, padding: $(2,2)$ |
| | BatchNorm2d $\rightarrow$ ReLU() |
| Conv2d | kernels: 64, kernelSize: $(5,5)$, stride: $(2,2)$, padding: $(2,2)$ |
| | BatchNorm2d $\rightarrow$ ReLU() |
| Conv2d | kernels: 128, kernelSize: $(5,5)$, stride: $(2,2)$, padding: $(2,2)$ |
| | BatchNorm2d $\rightarrow$ ReLU() |
| Linear | in: $(c \times w \times h)$, out: 4096 |
| | BatchNorm1d $\rightarrow$ ReLU() |
| Linear | in: 4096, out: 256 |
| | BatchNorm1d $\rightarrow$ ReLU() |

Table 6: Encoder network MNIST

| Decoder Architecture MNIST | |
| --- | --- |
| Linear | in: 20, out: 4096 |
| | BatchNorm1d $\rightarrow$ ReLU() |
| Linear | in: 4096, out: $(c \times w \times h)$ |
| | BatchNorm1d $\rightarrow$ ReLU() $\rightarrow$ Rearrange('b (c w h) -> b c w h') |
| ConvTranspose2d | kernels: 128, kernelSize: $(3,3)$, stride: $(1,1)$, padding: $(0,0)$ |
| | BatchNorm2d $\rightarrow$ ReLU() |
| ConvTranspose2d | kernels: 64, kernelSize: $(5,5)$, stride: $(2,2)$, padding: $(1,1)$ |
| | BatchNorm2d $\rightarrow$ ReLU() |
| ConvTranspose2d | kernels: 32, kernelSize: $(5,5)$, stride: $(2,2)$, padding: $(1,1)$ |
| | BatchNorm2d $\rightarrow$ ReLU() |
| ConvTranspose2d | kernels: 1, kernelSize: $(5,5)$, stride: $(1,1)$, padding: $(2,2)$ |
| | BatchNorm2d $\rightarrow$ Sigmoid() |

Table 7: Decoder network MNIST

| Motion Network MNIST | | | |
| --- | --- | --- | --- |
| Position | | Momentum | |
| Linear | in: 256, out: 320 | Linear | in: 256, out: 320 |
| BatchNorm1d $\rightarrow$ LeakyReLU(0.2) | | BatchNorm1d $\rightarrow$ LeakyReLU(0.2) | |
| Linear | in: 320, out: 20 | Linear | in: 320, out: 20 |
| BatchNorm1d $\rightarrow$ LeakyReLU(0.2) | | BatchNorm1d $\rightarrow$ LeakyReLU(0.2) | |
| $\text{Linear}_\mu$ | in: 20, out: 20 | TCN | kernelSize: 4, pad: 3, stride: 1 |
| $\text{Linear}_{\log \sigma}$ | in: 20, out: 20 | $\text{Linear}_\mu$ | in: 20, out: 20 |
| | | $\text{Linear}_{\log \sigma}$ | in: 20, out: 20 |

Table 8: Motion network MNIST. TCN stands for temporal convolution network.

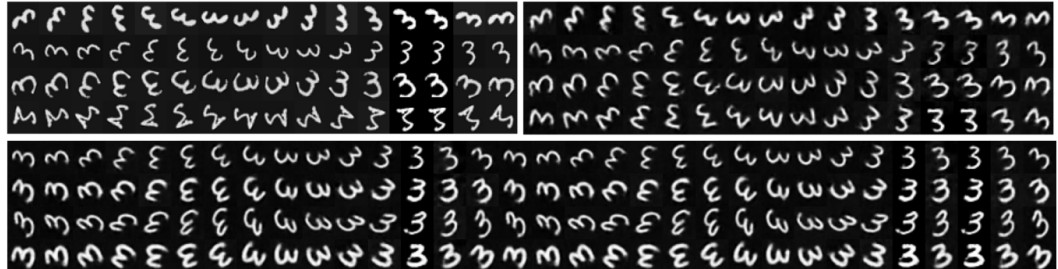

Figure 5: Results on Rotating MNIST with a learnable Hamiltonian operator. On the top left, we have four input sequences, and on the right, their reconstruction; on the bottom, we have four sequences generated by an action of Hamiltonian on the state space coordinate of the frame in the first column.

| Classifier Architecture | |
|---|---|
| Conv2d | kernels: 64, kernelSize: $(5,5)$, stride: $(4,4)$, padding: $(1,1)$ |
| | BatchNorm2d $\rightarrow$ LeakyReLU(0.2) |
| Conv2d | kernels: 128, kernelSize: $(5,5)$, stride: $(4,4)$, padding: $(1,1)$ |
| | BatchNorm2d $\rightarrow$ LeakyReLU(0.2) |
| Conv2d | kernels: 256, kernelSize: $(5,5)$, stride: $(4,4)$, padding: $(1,1)$ |
| | BatchNorm2d $\rightarrow$ LeakyReLU(0.2) |
| Linear | in: $(c \times w \times h)$, out: 1024 |
| | BatchNorm1d $\rightarrow$ LeakyReLU(0.2) |
| LSTM | in: 1024, out: 512 |
| | BatchNorm1d $\rightarrow$ LeakyReLU(0.2) |
| Linear | in: 512, out: 256 |
| | BatchNorm1d $\rightarrow$ LeakyReLU(0.2) |
| Linear | in: 256, out: $K$ |

Table 9: Classifier network used for evaluation. For the attribute classification task, $K$ is set to the number of attributes and for the action classification, it is set to the number of actions.

Easton, R. W. (1993). Introduction to Hamiltonian dynamical systems and the N-body problem (KR Meyer and GR Hall). *SIAM Review*, 35(4):659–659.

Kingma, D. P. and Ba, J. (2014). Adam: A method for stochastic optimization. *arXiv preprint arXiv:1412.6980*.

Paszke, A., Gross, S., Massa, F., Lerer, A., Bradbury, J., Chanan, G., Killeen, T., Lin, Z., Gimelshein, N., Antiga, L., Desmaison, A., Kopf, A., Yang, E., DeVito, Z., Raison, M., Tejani, A., Chilamkurthy, S., Steiner, B., Fang, L., Bai, J., and Chintala, S. (2019). Pytorch: An imperative style, high-performance deep learning library. In Wallach, H., Larochelle, H., Beygelzimer, A., d'Alché-Buc, F., Fox, E., and Garnett, R., editors, *Advances in Neural Information Processing Systems 32*, pages 8024–8035. Curran Associates, Inc.

Yildiz, C., Heinonen, M., and Lähdesmäki, H. (2019). ODE2VAE: Deep generative second order ODEs with Bayesian neural networks.

Yingzhen, L. and Mandt, S. (2018). Disentangled sequential autoencoder. In *International Conference on Machine Learning*, pages 5670–5679. PMLR.

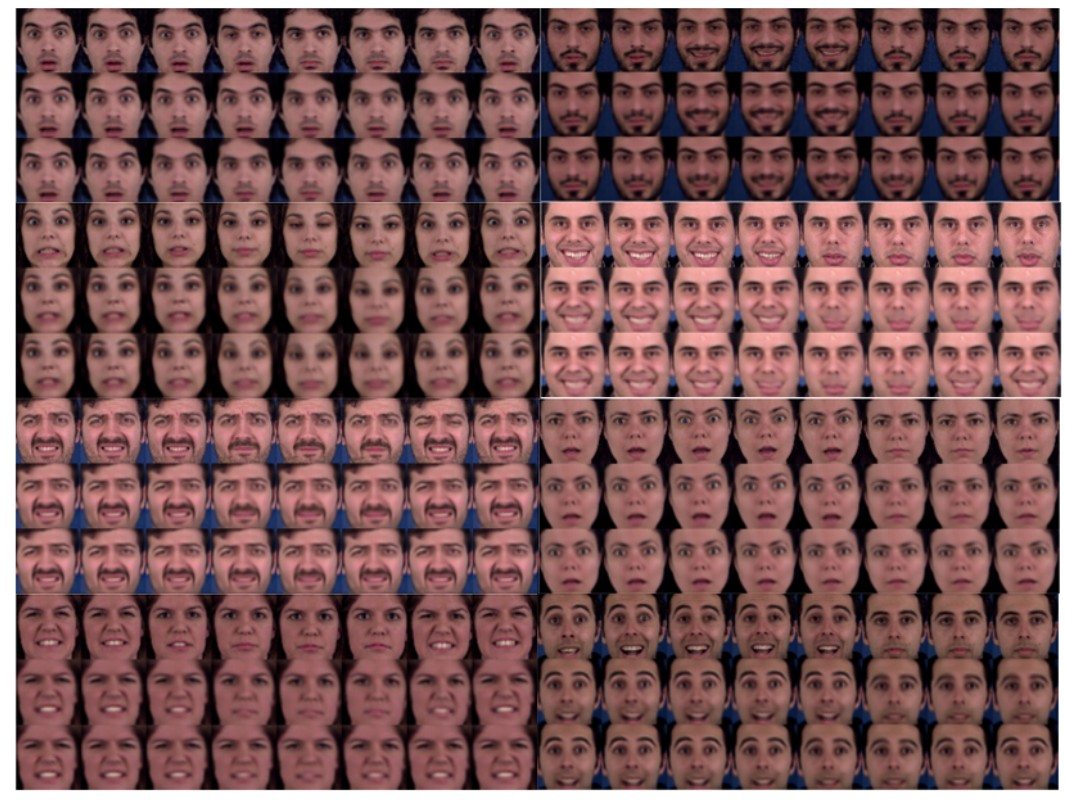

(a) Conditional Sequence Generation. The first row is the original sequence, the second row is a reconstructed sequence, and the third is generated by an action of a dynamical model on the first time frame

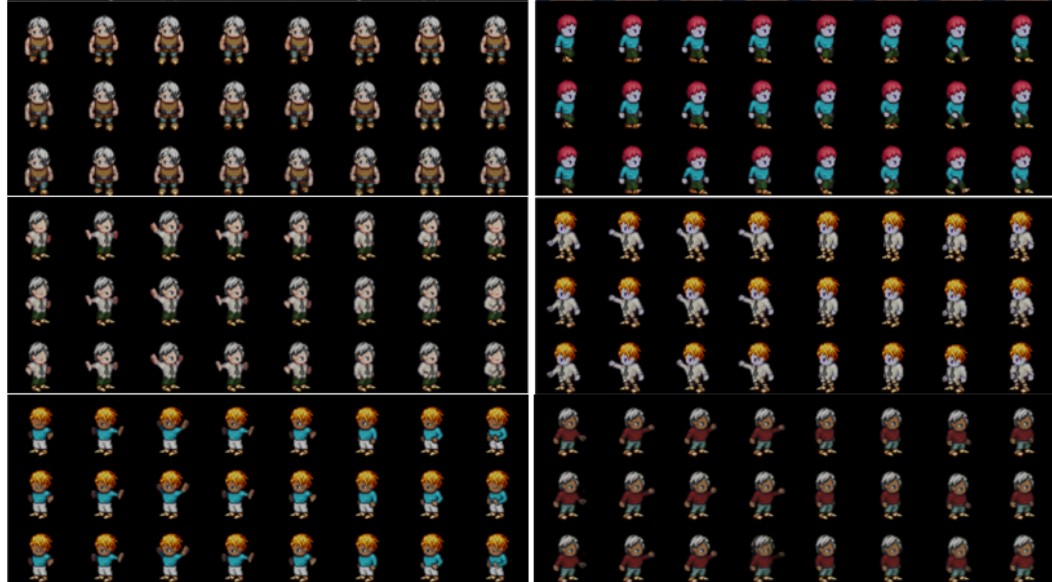

Figure 6: Conditional Sequence Generation. The first row is the original sequence, the second row is a reconstructed sequence, and third is generated by an action of a dynamical model on the first time frame

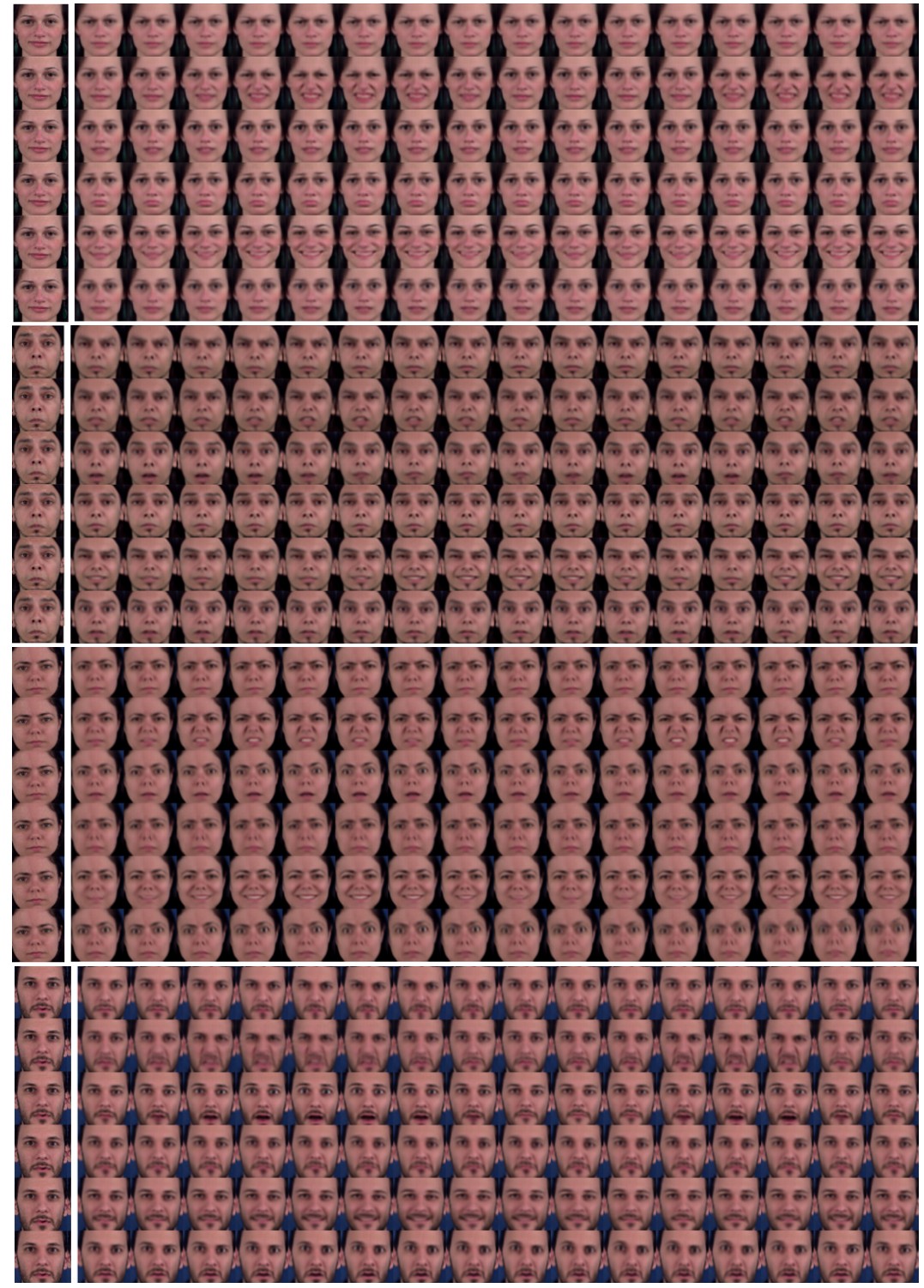

Figure 7: Image to Sequence generation. We generate dynamics of different actions from a given image. Each row is a unique action generated by the operator associated with that action.

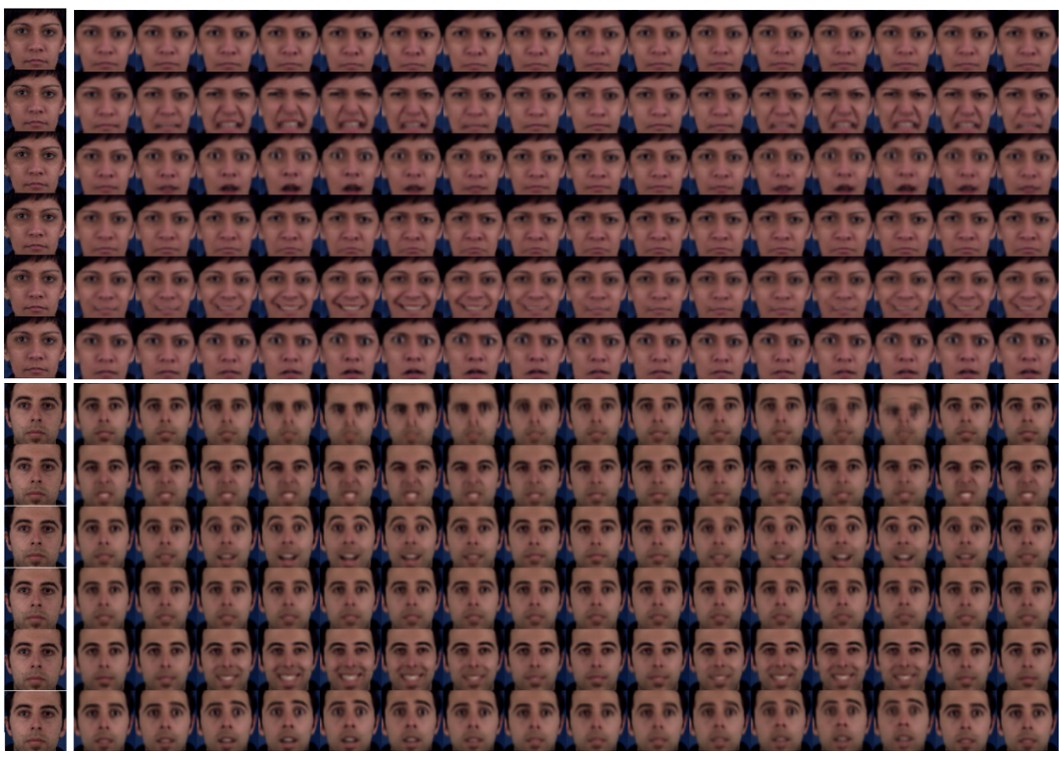

Figure 8: Image to Sequence generation. We generate dynamics of different actions from a given image. Each row is a unique action generated by the operator associated with that action.

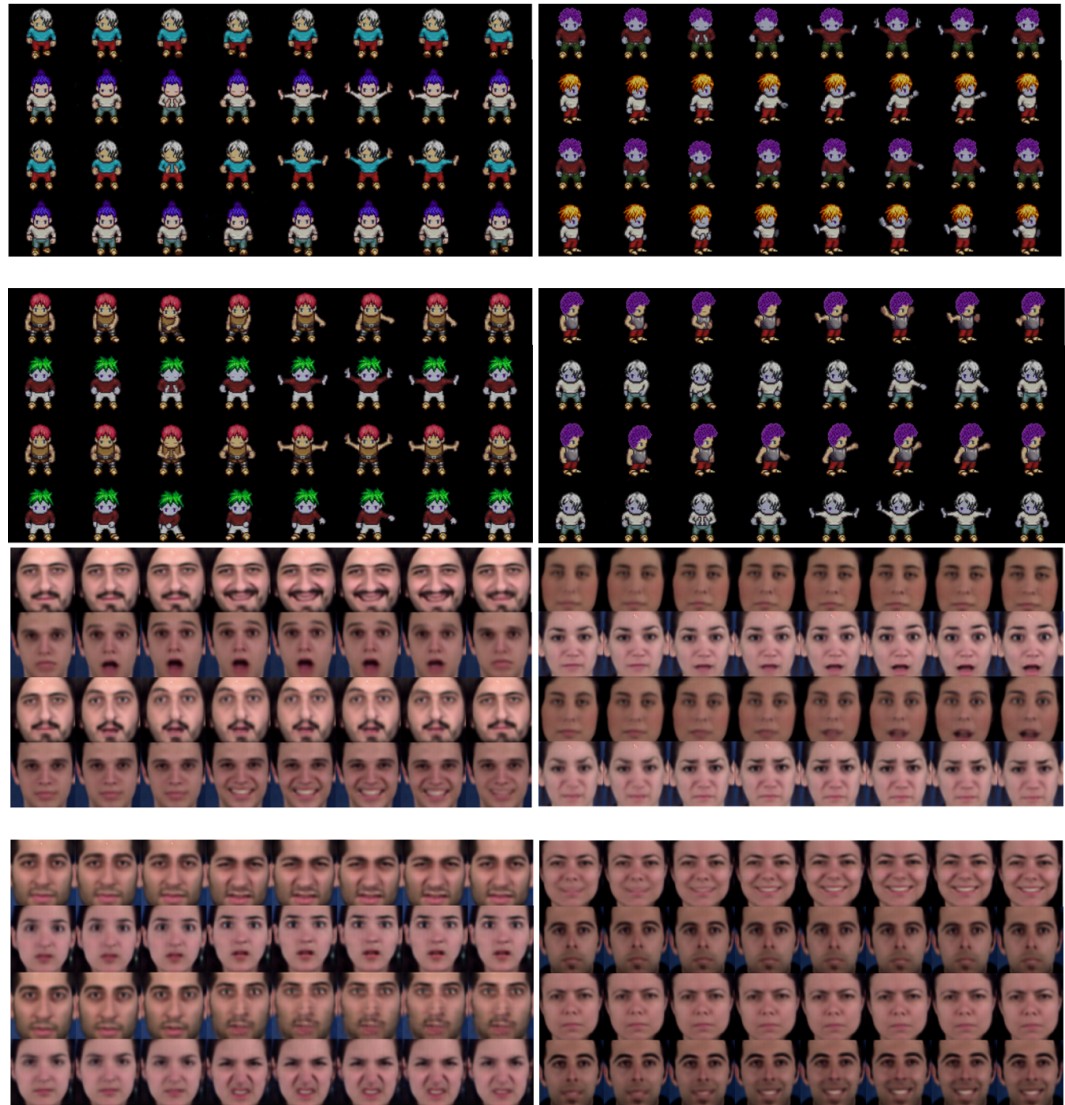

Figure 9: Motion Swapping. In each patch, the first two rows are the original sequence and the next two rows are obtained by swapping motion variables of two sequences.