# OpenReview forum: "Hamiltonian Latent Operators for content and motion disentanglement in image sequences"
_NeurIPS.cc/2022/Conference — NeurIPS 2022 Accept_

### Official Review · Reviewer_FnRf · 2022-07-07

**Rating:** 6
**Confidence:** 3
**Soundness:** 3 good
**Presentation:** 3 good
**Contribution:** 3 good

**Summary:**

This paper deals with the task of generating image sequences. Specifically, the authors propose a method called Halo that allows to disentangle the content from the motion in image sequences, in the VAE framework. They do so by separating the latent space in two spaces: 1) the content space, a global content vector that summarizes the image sequence;  2) the motion space, a sequence of time-dependent vectors that capture the dynamics of the sequence. The main contribution of the authors is to model the motion space with Hamiltonian dynamics. The authors claim that Hamiltonian dynamics have good inductive biases for sequence generation, such as reversibility of the motion. Experiments on simple image sequences are performed to prove the quality of their model.

**Questions:**

1) Are $H_k$ matrices learned, or are they kept as drawn from the initialization with symmetric matrix $M_k$ and $H_k = J M_k$. If learned, how  is $H_k$ constrained during optimization?
2) Does the inductive bias of the Hamiltonian operator help in a setting with few training examples?

**Limitations:**

The authors addressed the limitations.

**Strengths And Weaknesses:**

Strengths
* 1) The latent Hamiltonian operator is quite generic. It could be extended and used with other families of deep generative models for sequences, and thus be of great interest for practitioners.
* 2) Halo achieves SOTA scores on motion/content disentanglement metrics. Ablations with similar architecture and other sequence models are convincing (especially Table 1 of Supplementary Material).


Weaknesses
* 1) Stochasticity is only allowed by the gaussian sampling, but there is no stochasticity in the Hamiltonian operator. Thus, Halo can only generate one motion vector given input frames. However, trajectory prediction is a highly stochastic process. This could be a limitation that makes scaling to more complex environments difficult.
* 2) How does the method guarantee that no motion information leak in the global content vector? Since the encoder + LSTM that generate the global vector see the whole input sequence, it could also capture some information about the motion.
* 3) Limited evaluation: the model is tested on three simple datasets. Could be interesting to see how it performs on more complex datasets, with different/moving backgrounds.
It also misses comparisons with recent works. E.g.:
a) Franceschi et al., "Stochastic latent residual video prediction". ICML 2020.
b) Wang et al., "G3AN: Disentangling appearance and motion for video generation". CVPR 2020.
* 4) Clarity. While the paper is well written, there lacks some implementation details on the main component of the paper: the hamiltonian operator (see questions). This affects the understandability.

---

> ### Author Response · Authors · 2022-08-02
> **Response to Reviewer FnRf**
>
> We thank the reviewer for their valuable assessment of our work. We are glad our work interests you. Our response to your concerns are below:
>
> i) ***Clarification on learning $H_k$ matrices***
>
> The Hamiltonian operator $H=JM$, where $J$ is a fixed matrix as stated in definition  1. The matrix $M$ is parameterised as a learnable symmetric-matrix as $M = 0.5(A+A^T)$, where entries in $A$ are learnable real-valued parameters. Many thanks for your comments. We have updated it in the paper. We will rectify any further clarification needed in the final version.
>
> ii) ***Sample efficiency of Hamiltonian inductive bias***
>
> Indeed, sample efficiency is an interesting question. In our current work, we did not investigate it. We consider it will be interesting to examine the applicability of Hamiltonian operators in model-based reinforcement learning where an environment is composed of various complex physical dynamics. Hamiltonian inductive bias can help learn a model of the world in such scenarios. We consider it as the future scope of the work.
>
>
> iii) ***Handling complex stochastic videos***
>
> Given the initial coordinate $(p_0, q_0)$ and its Hamiltonian energy, Hamilton's equation uniquely determines the path in phase space. For highly stochastic dynamics can introduce a mechanism to inject/remove energy from the system. This can be possible, for instance, by making $H$ dependent on time. But this would imply a need for a more sophisticated integration scheme. Thank you for pointing out an interesting question. We indeed consider this as a potential limitation in the current formulation and would like to address it in future work.
>
>
> iv) ***How does the method guarantee that no motion information leaks in the global content vector?***
>
> The point of our model is to capture the content and motion across multiple different sequences with different motions. The neural network is shared between these sequences, and the only information about any individual image in the sequence or the dynamic process is in the representation. Indeed the neural decoder will capture information that is common across all frames in all images. However, because it is common for each frame it cannot capture any motion in itself. Furthermore, because the content part is the same for all parts of the motion, it too cannot contain information that distinguishes the current motion position information. The optimisation does nothing in these terms. This decomposition is enforced by the representational structure itself.
>
>
> The global vector $z$ is a summary statistics of a sequence. A naive way would be to take an average of the frame level encodings. We used LSTM for a fair comparison as it was used in the baseline DSVAE paper. For any two timesteps, say 0 and 1, let the respective motion components be $q_0$ and $q_1$. Then the combined latent coordinates as an input to the decoder are $[z,q_0]$ and $[z, q_1]$. For the decoder the right temporal information in $x_0$ and $x_1$ can only come from $q_0$ and $q_1$ as $z$ is common. Thus preventing any motion from slipping to the $z$ space. Furthermore, the symplectic form (due to matrix $J$) ensures the volume element is preserved in the phase space. As a result, the vector field in phase space has a zero divergence, implying there are no points where dynamics can converge/diverge, ensuring no static information in phase space.
>
> v) ***References to the related work***
>
>  Thank you for pointing out the related paper. In comparison, we included methods with similar architecture and parameters. We will include the additional references in the final version. Our main objective was to demonstrate a formulation that can leverage Hamiltonian structure for disentangling content and motion. We indeed agree would be interesting to investigate the application in challenging scenarios with multiple moving objects. We appreciate the suggestions and would like to generalise our approach in future work.

---

> ### Author Response · Authors · 2022-08-09
> **Follow up**
>
> Thank you for your valuable feedback on our paper. We really appreciate your positive comments. To the best of our effort, we responded to your reviews. As the discussion period is ending soon, we are wondering whether our responses helped address your concerns. Please let us know if you need any further clarification; we will be happy to provide further details.

---

### Official Review · Reviewer_pzKP · 2022-07-10

**Rating:** 5
**Confidence:** 3
**Soundness:** 2 fair
**Presentation:** 3 good
**Contribution:** 3 good

**Summary:**

The paper proposes Halo, a novel type of variational autoencoder with structured latent space and demonstrates its applications to different types of (controlled) video generation tasks. The main contribution is a principled decomposition of the latent space into a content space and a motion space, where the motion space is modeled using Hamiltonian dynamics. The structural constraints (e.g, symmetries) imposed by these dynamics induce desirable properties like reversibility and volume-preservation, enabling the conservation of (learned) quantities.

**Questions:**

- l.201f: *“without significant loss of generality, we propose a linear Hamiltonian system in the latent layer, relying on the deep neural network mapping to data space to handle all nonlinear aspects.”* This comment raises a larger question: how does the optimization process ensure that not *all* relevant information is pushed to the decoder, while the carefully designed latent space models only trivial dependencies? In the same vein, how does the optimization process guarantee that no motion data is being pushed into ${\bf z}$, thus completely circumventing the phase space?

- It is not entirely clear to me how the Hamiltonian nature of ${\bf H}_k$ is preserved during the optimization process. My understanding is that the symmetry of ${\bf M}_k$ ensures this, but I would appreciate a confirmation.

- l.224: *“The symplectic geometry proves useful for long term term *(sic!)* sequence generation.”* The sequences in the main paper are all relatively short and even the experiments on longer sequences shown in the Appendix are only 32 frames, i.e., a little over a second at 25fps. What is the limit before the generated sequences visibly deteriorate or become static?

- What architecture does the pre-trained action prediction classifier use and how was it trained?

**Limitations:**

- The paper flags potential misuse in the area of fake video data generation.
- The paper does not contain a limitations section.

**Strengths And Weaknesses:**

**Strengths**
+ Video generation is an important and challenging task with a rich history. The proposed approach takes a fresh perspective on this topic and explores the benefits of inductive bias based on principles rooted in the physics community.

+ Sections 1-3 (introduction, related work, method) are well-organized and easy to follow: the main contributions are clearly formulated, the figures are helpful in understanding architectural details, and the mathematical notation is (mostly) consistent. The related work section is commendable and provides a comprehensive overview of the field. The paper does have a fairly strong physics flavour and I would recommend to provide stronger guidance for an audience which may not be familiar with topics that are not part of core ML, such as group action, phase space, conservation law, and symplectic geometry.

+ The Hamiltonian design of the latent space is interesting and novel, and the advantages of reversible and symplectic latent dynamics make intuitive sense. I also appreciate the principled derivation of the dynamical model $f$ from a constant-energy perspective (l.206-l.214). The variational inference section is less clear and I would encourage the authors to move at least some intuition about the ELBO from the Appendix into the main paper.

**Weaknesses**

- The weakest part of the paper are its experiments, both in terms of their presentation and design.
	- Presentation:
		- The structure of the experiments is confusing throughout section 4. For example, the description of the Sprites and MUG datasets starts in the middle of the “Rotating Balls” paragraph (l.262). Likewise, the description of the baseline comparison starts in the middle of the “Quantitative Evaluation” section (l.302). Grammar and text flow in the experiment section also feel unpolished. Finally, none of the figures in this section have proper axes/labels and the reader needs to count rows and infer the content from the caption or even the main text (l.348-350 for Figure 5 (left)).

	- Design:
		- Since Table 1 does not include a comparison to other baselines it is not possible to assess whether the presented SSIM/PSNR/MSE scores are competitive or not. Why not use the same metrics as in Table 2?
		- Table 3 is not mentioned in the text and seems to be based on the single example of Figure 5 (right), which is not enough to make any general statements. The positional encoding mentioned in this table is not explained and not supported by any qualitative evidence.
		- In Figure 4 (left) the reconstructed and generated sequences look fairly similar, which can be an indication of low diversity.
		- It is unclear how the sequences of the rotating balls dataset were generated as the mentioned constraint does not specify any temporal pattern. What is the dynamic model used here?
		- The sequences are very short (8/16 frames) and small (64 x 64). What is the main bottleneck that prevents application to high-fidelity image sequences?

**Minor comments**

- Typos: Figure 1 (“alongwith”), l.224 (“long term term”), l.234 (“(6))”), l.258 (as”blue”), l.283 (“EvaluationWe”)

- The Appendix provides valuable information about the ELBO objective, terminology, and network structures, but the main paper does not refer to it often enough (e.g., content/position/momentum network).

- The paper follows a top-down approach, first introducing high-level structures and then filling in the details. While that is a reasonable approach, it does mean that readers will have to read the paper twice (or go back to previous paragraphs), because the motivation for some design choices remains initially unclear. One example is the structure of the phase space.

**Summary**. I appreciate the technical formulation of this paper but am on the fence due to the weak and unconvincing experiments. I encourage the authors to address the concerns above as well as the questions below.

---

> ### Author Response · Authors · 2022-08-01
> **Response to Reviewer pzKP (Part 1)**
>
> We thank the reviewer for their valuable assessment of our work. Our responses to your concerns are below:
>
> i) **Text flow in results**
>
> Thank you for your helpful comments on the presentation. We have polished the results section per your feedback and fixed the typos.
>
> ii) **How does the optimization process ensure that not all relevant information is pushed to the decoder**
>
> The point of our model is to capture the content and motion across multiple different sequences with different motions. The neural network is shared between these sequences, and the only information about any individual image in the sequence or the dynamic process is in the representation. Indeed the neural decoder will capture information that is common across all frames in all images. However, because it is common for each frame it cannot capture any motion in itself. Furthermore, because the content part is the same for all parts of the motion, it too cannot contain information that distinguishes the current motion position information. The optimisation does nothing in these terms. This decomposition is enforced by the representational structure itself.
>
> Specifically, consider global $z$ shared across all timesteps. For any two timesteps, say 0 and 1, let the respective motion components be $q_0$ and $q_1$. Then the combined latent coordinates as an input to the decoder are $[z,q_0]$ and $[z, q_1]$. For the decoder the right temporal information in $x_0$ and $x_1$ can only come from $q_0$ and $q_1$ as $z$ is common. Thus preventing any motion from slipping to the $z$ space. Furthermore, the symplectic form (due to matrix $J$) ensures the volume element is preserved in the phase space. As a result, the vector field in phase space has a zero divergence, implying there are no points where dynamics can converge/diverge, ensuring no static information in phase space.
>
>
> iii) **How is the Hamiltonian structure preserved**
>
> Indeed, the matrix $M$ and $J$ ensure the Hamiltonian nature of the latent space. Specifically, the Hamiltonian operator $H=JM$, where $J$ is a fixed matrix as stated in definition  1. The matrix $M$ is parameterised as a learnable symmetric-matrix as $M = 0.5(A+A^T)$, where entries in $A$ are learnable real-valued parameters. Many thanks for your comments. We have made updates to the paper. We will rectify any further clarification in the final version.
>
>
> iv) **Design of results in Table 1**
>
> We want to clarify the central theme of our paper is using Hamiltonian operators for disentanglement of motion from content variables. We do a comparison with baseline methods on the disentanglement task in Table 2. Table 1 investigates the effect of different geometry by evaluating the reconstructed and predicted sequences. The Hamiltonian operators give rise to a symplectic geometry in the latent space. Introducing further restrictions on the matrix $H$ results in a different geometry in phase space; for instance, restricting to skew-$H$ will confine the operator to rotations which are easy to interpret. For this purpose, we sampled a starting step from a ground-truth sequence and predicted the future trajectory, compared against the available ground truth. We observe that imposing extra constraints on $H$ did not improve perceptual scores. For the rest of the experiments, we only consider $H$ and evaluate commonly used disentanglement metrics as done in baseline methods.
>
> Indeed the evaluation can be done directly on the disentanglement task using evaluation metrics similar to Table 2. We chose to look at perceptual scores and select the best operator so our central disentanglement evaluations could be more concise, especially the number of qualitative figures added to the paper.
>
> v) **Missing reference to positional encoding**
>
> Due to limited space, we discuss the positional encodings in appendix Section 3.3. We have added a reference in the main paper.
>
> vi) **Diversity of sequence and clarification on Figure 4**
>
> In Figure 4, we compare the original reconstruction and the generated sequences. We use the first two timesteps to obtain the initial position and momentum coordinates and unroll the trajectory in latent space. Since the momentum is determined from previous frames, we can compare the unrolled trajectory with the known target trajectory. For generating diverse sequences, we can sample different initial momentum variables in latent space and combine that with position to unroll the trajectory. We do that in the image to sequence results.

---

> > ### Comment · Reviewer_pzKP · 2022-08-07
> > **Helpful clarifications**
> >
> > I appreciate the updated manuscript and detailed response. I can see an improvement in some of the areas pointed out in my original review (e.g., structure/clarity of presentation) and the provided clarifications about motion/content separation, sequence length/deterioration, and the underlying datasets are helpful. I would suggest to include parts of this response in the paper as well. The rebuttal does not fundamentally change my view of the paper and I still believe that HALO is a technically interesting approach with weaknesses in its experimental validation; however, I do appreciate the additional context provided by the authors. I feel compelling experiments on less constrained data (Sprites is a very controlled setting and even the evaluation on real-world face data is still in the realm of toy data given near-perfect alignment and orientation) could make this a much stronger paper. I’m not opposed to acceptance but also do not see changes to the manuscript that are significant enough to raise my initial rating.

---

> > > ### Author Response · Authors · 2022-08-07
> > > **Further updates**
> > >
> > > Thank you for your valuable feedback. We included your helpful suggestions for improving the presentation of the results. As the rebuttal phase has a 9-page limit, we have noted the suggestions and will incorporate all the rebuttal phase's clarifications in the final version of the paper. The dataset we selected for the experiments was employed in the baseline methods DSVAE, S3VAE, etc. We indeed agree it would be more interesting to look at challenging real-world problems. In this work, we wanted to demonstrate the Hamiltonian formulation provides a more principled way to think about content and motion disentanglement. The Hamiltonian formulation has many attractive properties such as reversibility of dynamics, symplectic structure in latent space and a bilinear form of energy. We demonstrated the benefits of these properties on commonly used datasets. In our appendix, we conducted extensive ablation studies to study the effect of constant energy in the motion space.
> > > Furthermore, we want to note our dynamical model is linear, which makes it easy to interpret (unlike existing approaches using non-linear models). The block-diagonal structure of Hamiltonian makes it easy to scale. We hope you consider our contribution in your final rating. We will be happy to address any remaining concerns in our existing validation. Extending to more complex data comes with an additional computational cost and challenges, such as handling injection/removal of energy in stochastic environments. We note these more exciting applications and will definitely extend our framework to such scenarios in future work.

---

> > > > ### Author Response · Authors · 2022-08-09
> > > > **Follow up**
> > > >
> > > > We are very thankful for your detailed feedback on the paper and for responding to the rebuttal. We appreciate it a lot. We replied to your reviews to the best of our effort and promise to incorporate feedback in the final version. Due to limited time and computational issues pointed out in our rebuttal, we consider more complex data scenarios as the future scope of the work. Please let us know if you have any other remaining concerns or need clarification; we will be happy to provide further details. Thanks again!

---

> ### Author Response · Authors · 2022-08-02
> **Response to Reviewer pzKP (Part 2)**
>
> vii) **Dynamics in rotating balls dataset**
>
> For the rotating balls experiment, we restrict the centre of a ball in sequences to move along an orbit of a fixed radius $c$ from the centre of a frame. We first sample an initial frame containing a ball whose $(x,y)$ centre lies in orbit. Next, the future steps are drawn by rotating the center of a ball in an anticlockwise direction that is sample $T$ angles in $[0, 2\pi]$ and place $(x,y)$ center of ball at $(c\cos (\theta), c \sin (\theta))$. We generate multiple sequences by adding a small random noise to the initial location and varying the radius $c$. This dataset can be viewed as a pendulum with a known conserved quantity. The results on the toy dataset demonstrate we can use our approach to swap the content of sequences rotating with a different conserved energy.
>
> viii) **Length of sequences used. Scalability to higher resolution images**
>
> The length of sequences used for training and evaluation is chosen to be consistent with the baseline comparison methods. As you observed in Appendix Figure 3, we demonstrate longer length sequences $32$. The computational resources are the main bottleneck in the training and evaluation high-resolution sequences. In each training batch, the model's input is of size $B\times T\times C\times W\times H$. A large value of $T$, $W$ and $H$ increase the cost of storing intermediate network outputs and the model. We think our approach can be scaled up for high-resolution images by incorporating more sophisticated encoder-decoder architectures and using large GPUs. We want to remark our main claim is to demonstrate the benefits of symplectic geometry for disentangling motion from content in image sequences. In this work, we present the results on three commonly used datasets. Our model is more general and can be combined with other developments on scaling VAEs to high-fidelity images.
>
>
> ix) ***Deterioration in decoder output***
>
> Our experiments did not observe deterioration even when used for long trajectory prediction. As also demonstrated in Figure 3, the overall energy stays constant over time. We use an explicit solution of a dynamical system in the form of matrix exponential to unroll the trajectories, unlike other models where dynamics tend to deviate from the data manifold due to the accumulation of error over timesteps. This accumulation error results in visual distortions or dynamics' static nature. However, this is not the case in our formulation. We also remark the deterioration in images does appear when sampling a new content variable from the prior in a latent space and using that with the motion variables to unroll the trajectory. This effect is due to the mismatch between aggregated posterior and prior. Consequently, there are regions in latent space with low density. Such issues can be addressed using a $\beta$ formulation of VAE and carefully tuning the $\beta$ parameter.
>
>
> x) ***Details of pretrained classifier***
>
> We have provided architecture details of a pretrained classifier in the Appendix with references in the main paper. We have also fixed the typos. Due to limited space, we couldn't include further details on ELBO in the main paper at this point. We promise to include intuition of ELBO in the final version and address any further readability issues.

---

### Official Review · Reviewer_pj74 · 2022-07-17

**Rating:** 5
**Confidence:** 4
**Soundness:** 2 fair
**Presentation:** 3 good
**Contribution:** 3 good

**Summary:**

This paper proposes a deep state space model for videos. The dynamics are defined by linear Hamiltonian Dynamics, and the motion matrix is further assumed to be block diagonal in order to separate different categories of actions. Like previous works, a latent variable z is introduced for explaining content and kept fixed for all frames. Experiments are carried out on Sprites and MUG to demonstrate the efficacy.

**Questions:**

Please see comments above.

**Limitations:**

Yes.

**Strengths And Weaknesses:**

Strengths:

The paper is well written and easy to follow. The idea of introducing Hamiltonian dynamics as an inductive bias for explaining repetitive or cycled motions in videos are reasonable and natural. The theoretical derivation is technically sound.

Weaknesses:

My main concern is that the proposed method is not well supported by the experimental results:
- I don't understand how SSIM and PSNR can be used for evaluating "generation quality", as generated samples are supposed to be different from the training datasets. I can only imagine that the numbers in Table 1 are reported for reconstruction, in which case it is not for generation quality as described by the paper. Also, no baseline methods are compared in terms of reconstruction.
- There's no commonly used metrics reported that are designed for really evaluating sample quality, such as FVD or Inception scores.
- For disentanglement evaluation, it is not fair to use the conditional Halo model to compared with the baselines which are trained unconditionally. The only fair way is the compare unconditional Halo models with the baselines where the performance of the proposed model does not stand out.
- In the ablation study section, it makes me confused that the paper mentioned RNN or linear dynamic model cannot make image move but it also showed the results of swapping motions of the two baselines where the video sequences are changing over time. Also it sounds wired to me that the linear/RNN dynamics cannot do image-to-seq as those have been applied by many classical state space models.

In experiments, two operators H and skew-H are compared but there's no official definition of skew-H in the previous sections.

The model assumes that the action space can be divided into subspaces where each subspace represents a unique action. This representation can be highly ineffective if the number of actions goes huge.

---

> ### Author Response · Authors · 2022-08-01
> **Response to Reviewer pj74**
>
> We thank the reviewer for their valuable assessment of our work. Our responses to your concerns are below:
>
> i) **SSIM and PSNR in Table 1**
>
> We want to clarify the central theme of our paper is using Hamiltonian operators for disentanglement of motion from content variables. We do a comparison with baseline methods on the disentanglement task in Table 2. Table 1 investigates the effect of different geometry by evaluating the reconstructed and predicted sequences. The Hamiltonian operators give rise to a symplectic geometry in the latent space. Introducing further restrictions on the matrix $H$ results in a different geometry in phase space; for instance, restricting to skew-$H$ will confine the operator to rotations which are easy to interpret. For this purpose, we sampled a starting step from a ground-truth sequence and predicted the future trajectory, compared against the available ground truth. We observe that imposing extra constraints on $H$ did not improve perceptual scores. For the rest of the experiments, we only consider $H$ and evaluate commonly used disentanglement metrics as done in baseline methods.
>
> ii) **Choice of evaluation metrics**
>
> The choice of disentanglement metrics is based on their use in various baseline methods: DSVAE, S3VAE, and MoCoGAN. We want to remark that intra-entropy and inter-entropy are fairly more informative and, when combined, are equivalent to the Inception score. The log of inception score is equal to the mutual information between variable $y$ and $x$ [1]. Specifically, $\log (IS) = MI(y;x)$, the $ MI(y;x) = H(y) - H(y|x)$ Using this we can write $IS=e^{(H(y) - H(y|x))}$, where $H(y)$ is an inter-entropy and $H(y|x)$ is an intra-entropy term reported in the paper. Looking at two scores provides a better view of generated samples. We can add IS in the final version.
>
> iii) **Comparison of conditional and unconditional models**
>
> We compare both conditional and unconditional versions of our model. The improvement in accuracy under Hamiltonian dynamics without
> incorporating action variables is 5\% over the best S3VAE,
> 11\% over MoCoGAN and 21\% over DSVAE, which is still significant over the baselines. We want to note that the Hamiltonian dynamical model is linear, making it simple and easy to interpret, which is not the case with other methods.
>
> iv) **Ablation Study**
>
> In the ablation study for the motion transfer, we swap the motion variables of two sequences obtained using an encoder network. This evaluation compares the representation of the encoder and doesn't evaluate the generative model of dynamics. In the qualitative valuation on image to sequence, we map an image to a latent space and use the RNN to unroll the future trajectory in the latent space where it seems not to work. We hypothesise this could be because RNN relies on a history of frames to predict the future. During training, the temporal structure of timesteps is helpful in learning representation. However, this failed on the image to sequence task as there is no history of frames to produce the dynamics.
>
> v) **Clarification on $H$ and skew-$H$ operators**
>
> By $H$ we refer to the Hamiltonian of the form $H=JM$ where $M$ is a symmetric matrix, and in skew-$H$, we further restrict $H$ to be a skew-symmetric matrix. We discuss it in lines 222-228 of the paper. We apologise for the lack of clarity. We have updated the description in Section 4.1.
>
>
> vi) **Handling large number of actions**
>
> In our formulation, the full Hamiltonian matrix takes the block-diagonal form where each block is a Hamiltonian of action of a respective subspace. The block-diagonal structure of the matrix makes it easy to parallelise and scale for a large number of actions. The knowledge of action space provides a valuable notion of disentanglement of a dataset with diverse dynamics. It offers the potential for exciting applications like (1) generation of controllable dynamics and (2) modelling complex motion as a composition of primitive motions. Our unconditional model shows such dependence is not a strict requirement.
>
>
> [1] Barrat, Shane and Rishi Sharma, "A note on the inception score."

---

> ### Author Response · Authors · 2022-08-09
> **Follow up**
>
> Thank you for your valuable feedback on our paper. To the best of our effort, we responded to your reviews. The discussion period is ending soon we are wondering whether our responses helped address your concerns. Please let us know if you need any further clarification; we will be happy to provide further details.

---

### Meta-Review · Area_Chair_okfX · 2022-09-06

**Recommendation:** Accept
**Confidence:** Certain

**Metareview:**

This paper proposes a novel type of variational auto encoder, referred to as HELO. The latent space is decomposed into a content space and a motion space, and the main contribution is the proposal to model the motion space using Hamiltonian dynamics. All reviewers agree that the idea of using Hamiltonian dynamics is interesting and novel. One main critique, that the authors agreed on, was that the operator does not contain any stochasticity and that this might be a limitation when applying the idea to model more complex data. Another remark was that the experiments are limited and experiments on less constraint data are missing. A quick look at the baseline methods revealed that they also use the same kind of data sets to evaluate their methods, so this latter concern might be of minor importance.
All in all, the potential positive outcomes of this paper outweight its current limitations, so we recommend acceptance at this point, while urging authors to address the remaining concerns in the final version.


**Award:**

No

---

### Decision · Program_Chairs · 2022-09-14

Accept